# DO PRETRAINED TRANSFORMERS REALLY LEARN IN-CONTEXT BY GRADIENT DESCENT?

## ABSTRACT

Is In-Context Learning (ICL) implicitly equivalent to Gradient Descent (GD)? Several recent works draw analogies between the dynamics of GD and the emergent behavior of ICL in large language models. However, these works make assumptions far from the realistic natural language setting in which language models are trained. Therefore, such discrepancies between theory and practice necessitate further investigation to validate their applicability.

We start by highlighting the assumptions in prior works that construct Transformer weights to simulate gradient descent. Their experiments with training Transformers on ICL objective, inconsistencies in the order sensitivity of ICL and GD, sparsity of the constructed weights, and sensitivity to parameter changes are some examples of mismatch from the real-world setting.

Furthermore, we probe and compare the ICL vs. GD hypothesis in a natural setting. We conduct comprehensive empirical analyses on language models pretrained on natural data (LLaMa-7B). Our comparisons on various performance metrics highlight the inconsistent behavior of ICL and GD as a function of various factors such as datasets, models, and the number of demonstrations. We observe that ICL and GD modify the output distribution of language models differently. These results indicate that the equivalence between ICL and GD is an open hypothesis, requires nuanced considerations, and calls for further studies.

## 1 INTRODUCTION

In-Context Learning (ICL) is an emergent behavior in Large Language Models (LLMs), which allows them to recognize patterns among demonstrations provided as prompts and extend these patterns to similar tasks (Brown et al., 2020). This fascinating on-the-fly learning behavior has motivated ample studies to better of understand its dynamics.

In particular, a notable line of work tries to explain ICL via Gradient Descent (GD) (Garg et al., 2022; Zhang et al., 2023). This connection is interesting because GD has been around for decades and is well-understood, while ICL is a recent phenomenon that has emerged somewhat surprisingly (Wei et al., 2022), and is not fully understood. Therefore, a solid formal bridge between the two approaches would be an exciting finding as it can open new doors for understanding ICL.

In this work, we revisit the hypothesis on the equivalence of ICL and GD, i.e., whether these two approaches to "learning"

> **Hypothesis 1.** For any Transformer weights resulting from self-supervised pretraining and for any well-defined task, ICL is algorithmically equivalent to $\overline{\text{GD}}$ (whole model or sub-model).

are functionally equivalent. Consider hypothesis 1 that defines a *universal* notion of equivalence between the ICL and GD. It defines equivalence as a property that must hold for *any* Transformer model with parameters that *emerge* naturally from pretraining on massive unlabeled data (Brown et al., 2020), and is applicable for *any* choice of well-defined tasks (Srivastava et al., 2023). For example, Dai et al. (2022) claims that ICL is equivalent to implicit finetuning.

However, other recent works have focused on a different claim outlined in hypothesis 2, which focuses on in-context

> **Hypothesis 2.** For a given well-defined task, there exist Transformer weights such that $\widehat{\text{ICL}}$ is algorithmically equivalent to GD (whole model or sub-model).

learning behavior that is **not emergent** (denoted as $\widehat{ICL}$). This deviates from hypothesis 1 in the family of models (differences in training setups) and family of tasks, as we will see in detail in §3. This hypothesis articulates a tangential target: being able to *simulate* GD on a given task with *some* (trained or hand-constructed) Transformer weights. Achieving this target is mainly concerned with the **expressivity** of Transformer architecture (Merrill et al., 2022; Chiang et al., 2023), ignoring how they may emerge from pre-training. A few notable works use this hypothesis to provide a theoretical argument for the ICL≈GD claim. Specifically, Akyürek et al. (2022); von Oswald et al. (2023) show (via a different set of arguments) that Transformer-based architectures (Vaswani et al., 2017), for appropriate choices of parameters, can process their in-context observations in a way that is equivalent to running gradient updates on an *implicit sub-model*'s parameters using the same demonstrations.

These claims are made under strong assumptions, which raises the question of whether these hold in practice or not. Specifically, **do the recent results focusing on hypothesis 2 provide any (even partial) evidence for hypothesis 1?** Although these works highlight interesting abilities of the Transformer architecture, their claims about the equivalence between ICL and GD are *too strong* for real-world models.

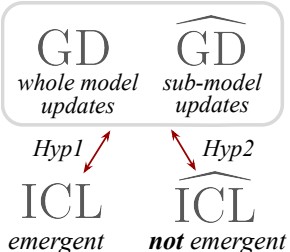

We divide our study into three parts. In the first part (§3), we show that previous works make assumptions in their study of the ICL≈GD hypothesis, that are hard to justify in the real world (hypothesis 2). Then, we use order-sensitivity as an argument against the equivalence between ICL and GD (§4). Finally, we put these claimed equivalences to the test (§5) by presenting a comprehensive empirical study. Our experiments reveal that ICL operates and performs differently from GD (fine-tuning the whole model or intuitive sub-models) on real-world language models across a variety of model sizes, datasets and the number of demonstrations.

In summary, (1) we provide theoretical and empirical arguments against existing theories regarding the equivalence of ICL and GD; (2) we empirically evaluate the equivalence between ICL and GD in the real-world setting using a variety of metrics and find that the two function quite differently; and (3) we highlight the gap in our understanding of the functional behavior of ICL between theoretical and real-world settings, and call for more nuanced and realistic studies.

## 2 BACKGROUND

We start with our problem setting (§2.1). We use "sampling" to emphasize a priori unknown problem parameters. Namely, sampling (choosing) a learning problem (task) and correspondingly sampling a pretrained model as the computational setup for our study. We then cover the two learning setups studied for equivalence (§2.2), followed by the treatment of ICL≈GD hypothesis in recent literature.

### 2.1 SAMPLING TASKS AND MODELS

**Sampling from the space of well-defined tasks.** Consider a family of functions (tasks) $\mathcal{F}$ such that each $(f : \mathcal{X} \rightarrow \mathcal{Y}) \in \mathcal{F}$, maps inputs in the domain $\mathcal{X}$ to the domain $\mathcal{Y}$. A particular function $f \in \mathcal{F}$ elicits a sampling process $x \overset{f}{\sim} \mathcal{X}$ which samples input from $\mathcal{X}$ such that they are compatible with $f$. For example, in natural language, $\mathcal{F}$ defines the space of all tasks that involve mapping from language input to language output, like sentence completion, summarization, QA, translation, etc. However, each task $f$ (e.g., translating English to French) would require specific inputs (English and not, say German) pertinent to the task. The goal is to find models that learn (imitate) $f$ by conditioning on a set of examples $S^f = \left\{ S_i^f = (x_i, f(x_i)) \middle| f \sim \mathcal{F}, x_i \overset{f}{\sim} \mathcal{X} \right\}$. The model's competence is then evaluated using a test set $S_{\text{test}}^f = \{(x_i^t, f(x_i^t))\}$, which is disjoint from $S^f$. During the evaluation, only the inputs in $S_{\text{test}}^f$ are passed to the model, which we denote as $X_{\text{test}}^f$.

**Sampling from the space of pretrained models.** LLMs like GPT and LLaMa (Brown et al., 2020; Touvron et al., 2023) are pretrained using the Causal Language Modelling (CLM) objective (Radford et al., 2019) which is more commonly understood as *next-word prediction* objective (Liu et al., 2018). This process of pretraining elicits a family of models $\mathcal{M}$ depending primarily on the *data*

*distribution and characteristics of sequences*, and additionally on the choice of architectures, initializations, etc. Formally, we denote this model $M_{\Theta_0}$ with pretrained weights $\Theta_0$, which is one model sampled from a much larger space of low perplexity pretrained models: $M_{\Theta_0} \sim \mathcal{M}$.

## 2.2 STANDARD LEARNING SETUPS

We review the standard treatment of ICL and GD and introduce the relevant notation.

**In-context learning (ICL).** We follow the dominant definition of In-context Learning (ICL) (Brown et al., 2020), which involves conditioning pretrained LLMs with a handful of examples of task $f$. Given these demonstrations, we want the LLM to perform $f$ on new inputs. Formally, given demonstrations $S^f = \{S_i^f\}_{i=1}^N$ and a test input $x_i^t \in X_{\text{test}}$, the model $M_{\Theta_0}$ generates a label $y_t$ when presented as $M_{\Theta_0}(S_1^f \circ S_2^f \circ ... S_N^f \circ x_i^t)$ or $M_{\Theta_0}(x_1 \circ f(x_1) \circ x_2 \circ f(x_2)...x_N \circ f(x_N) \circ x_i^t)$, where $\circ$ is a delimiter like new-line which separates the instances. $M_{\Theta_0}$ produces a confidence distribution $\in \mathbb{R}^{|V|}$ over the vocabulary set $V$.

**Gradient Descent (GD).** Gradient Descent is an iterative numerical optimization algorithm used to minimize a given objective with respect to model parameters. Given a model with initial parameters $\Theta_0$ and a differentiable loss function $\mathcal{J} \in \mathcal{Y} \times \mathcal{Y} \to \mathbb{R}$, the algorithm updates the parameters toward the negative gradient $\nabla_{\Theta_0} \mathcal{J}$. GD is a standard optimizer used to train neural networks including LLMs. Although there are variants, like SGD and Adam, that work well in practice, we focus our study on vanilla GD, which calculates the gradients and takes a step (learning rate $\eta$) of fixed size. In the context of learning from a set of demonstrations, pretrained models $M_{\Theta_0} \sim \mathcal{M}$ are fine-tuned on a particular task $f$ using GD by updating model parameters. Formally, parameter updates on the model $M_{\Theta_0}$ are performed for some epochs using the available demonstrations $S^f = \{S_i^f = (x_i, f(x_i))\}_{i=1}^N$ as follows:

$$\Theta_1 = \Theta_0 - \eta \nabla_\Theta \left( \frac{1}{N} \sum_{(x_i, f(x_i)) \in S^f} \mathcal{J}(M_{\Theta_0}(x_i), f(x_i)) \right). \tag{1}$$

After this process, the model is expected to perform this task given a new test sample *directly as input*: $M_{\Theta_1}(x_i^t)$.

## 3 THE LIMITING ASSUMPTIONS IN THE STUDY OF ICL≈GD HYPOTHESIS

We highlight how recent studies drift from these conventional definitions of ICL and GD (§2.2) to support another form of equivalence. Specifically, they put restrictive assumptions on both the space of models $\mathcal{M}$ and the space of tasks $\mathcal{F}$ when training Transformers. Additionally, they impose impractical assumptions on model weights needed to prove their notion of equivalence between ICL and GD. We discuss why these deviations from real practice are non-trivial and offer little support for the equivalence between ICL and GD in practical settings. Fig.1 encapsulates the theme of our arguments discussed in detail next.

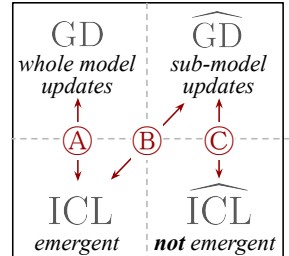

Figure 1: Ⓒ is discussed in §3. Ⓐ, Ⓑ in §4, §5;

### 3.1 REAL LLMS ARE NOT PRETRAINED WITH ICL OBJECTIVE

The widely-known ability of ICL *emerges* in pre-trained models ($\mathcal{M}$) that are obtained by training on CLM objective with natural language text as described in §2.1. Sequences in the pretraining corpus of natural language have a complicated relationship with the family of tasks $\mathcal{F}$ that they can perform using ICL. Understanding this relationship is an active area of research (cf. §6). However, we know that the pretraining corpus does not exclusively and explicitly contain sequences pertinent to $\mathcal{F}$. We refer to this training of Transformers with "natural" data (not necessarily natural language), which *does not* explicitly train it to perform ICL, as training with the *CLM objective*.

However, recent works use a different set of objectives. In (Akyürek et al., 2022; von Oswald et al., 2023; Garg et al., 2022), the models are trained using the *ICL objective*:

$$\arg\min_{\Theta} \mathop{\mathbb{E}}_{\substack{f \sim \widehat{\mathcal{F}} \\ x_i \overset{L}{\sim} \mathcal{X}}} \left[ \mathcal{L}\Big( f(x_i), M_{\Theta}(x_1 \circ f(x_1) \circ x_2 \circ f(x_2) \ldots \circ x_i) \Big) \right].$$

This deviates from the real settings in at least two aspects:

**Changing the space of tasks.** This objective is akin to training on the same restricted task distribution that the model is tested on via ICL. We call this $\widehat{\text{ICL}}$, or the ability to perform ICL by training on *ICL objective* (cf. Figure 1) and the corresponding family of tasks $\widehat{\mathcal{F}}$. For example, if the target task to learn is linear regression, the model is trained on the sequence of linear regression instances. Therefore, this setup does not necessarily capture the essence of how ICL *emerges* in LLMs, which are not trained to perform ICL on a family of tasks.

**Changing the space of models.** Moreover, optimizing for this objective elicits a family of models $\widehat{\mathcal{M}}$ that is embedded with the inductive bias of expecting a constant structure in the sequence: a series of $(x, y)$ pairs followed with a query input. Combined with the training on sequences specifically related to a restricted family of tasks $\widehat{\mathcal{F}}$, this space of models has different characteristics from the space of models $\mathcal{M}$ defined in §2.1.

The relationship between these sets of models is neither clear nor discussed in these recent works. Therefore, these works essentially equate $\widehat{\text{ICL}}$ with $\widehat{\text{GD}}$ (ⓒ in Figure 1). Although restricted to a stricter family of tasks like Linear Regression is reasonable for analysis, it is important to discuss these distinctions between the setups. Using the term *Transformers* to refer to both these spaces of models and using the term ICL for $\widehat{\text{ICL}}$ are both misleading.

## 3.2 HAND-CONSTRUCTED WEIGHTS AND THEIR LIMITS

In this section, we analyze the weight matrices constructed by von Oswald et al. (2023) and Akyürek et al. (2022). As no method is provided to arrive at these weights by training, we place these hand-constructed weights under the umbrella of $\widehat{\text{ICL}}$. Next, we show how they are hard to justify for real-world language models (e.g., LLaMa-7B).

We first re-write the weight matrices of Transformers constructed by von Oswald et al. (2023). Their proposition states that given a reference linear model $W$, there exist key, query, value, and projection matrices $(W_K, W_Q, W_V, P)$ of a Transformer such that a forward pass in that Transformer is identical to a gradient descent step on $W$, i.e., $e_j \leftarrow (x_j, y_j) + (0, -\Delta W x_j) = (x_i, y_i) + PVK^T q_j$.

The weight update $\Delta W$ is calculated by the mean squared error loss on the in-context samples as $\Delta W = -\eta \nabla_W L(W) = -\frac{\eta}{N} \sum_{i=1}^{N} (Wx_i - y_i)x_i^T$.

They construct $W_K = W_Q = \begin{pmatrix} I_x & 0 \\ 0 & 0 \end{pmatrix}, W_V = \begin{pmatrix} 0 & 0 \\ W_0 & -I_y \end{pmatrix}$ and $P = \frac{\eta}{N}I$, where $I_x, I_y$ and $I$ are identity matrices of size $N_x, N_y$ and $N_x + N_y$ respectively. Using these matrices, they achieve the dynamics of a gradient step in the forward pass of a Linear Self Attention Layer (without softmax). The construction by Akyürek et al. (2022) is more complex and requires multiple steps to simulate one step of GD on one in-context sample. However, the construction is similar in that it is similarly sparse (see section C.4 in Akyürek et al. (2022)'s appendix). These constructions raise multiple concerns about their scaling to real-world models.

**How does the model arrive at the correct $P$?** In the construction by von Oswald et al. (2023), $P$ is trivially assigned the value $\frac{\eta}{N}I$ which would change with the number of in-context samples. There is no insight into how a Transformer model would arrive at this information and how this formation behaves without any in-context samples. An edge case is $N = 0$ (no demonstrations), which surprisingly makes terms in $P$ go to infinity.

**Are LLM weights this sparse?** The weight construction by von Oswald et al. (2023) has a lot of extremely sparse weight matrices. To be precise, $W_K$ and $W_Q$ would be matrices with $N_x$ terms

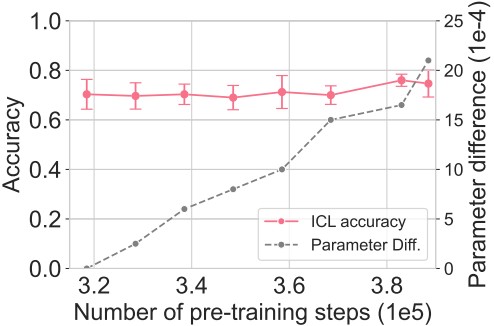
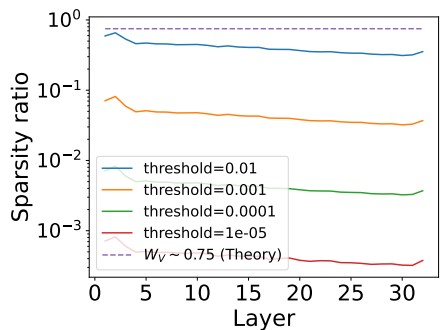

Figure 2: GPT-J's ICL ability does not change much over a time during training, while the parameters change steadily. 'Parameter difference' refers to the average parameter changes across $W_K, W_Q$, and $W_V$ over all layers. More results in Appendix B.

Figure 3: We show that the sparsity ratio $W_V$ in LLaMA is much less than previous works required to implement GD. More results are deferred to Appendix C.

equal to 1 in the top left of the diagonal with the rest of $(N_x + N_y)^2 - N_x$ terms equal to zero. For LLaMa, the embedding size of the token vector, $N_x = N_y = 4096$. This means that the sparsity ratio (SR) in the weight matrices should be $\frac{((N_x+N_y)^2 - N_x)}{(N_x+N_y)^2} > 99.99\%$. The sparsity ratio in $W_V$ should be close to $\approx 75\%$ if we assume each element in $W_0$ to be non-zero. In practice, the sparsity ratio is much lower for real-world models like LLaMa and GPT-J. As precisely 0 values for weights are unlikely, we measured the sparsity ratio in $W_K, W_Q$, and $W_V$ by measuring weights less than a threshold ($\delta$). Figure 3 shows the average sparsity value across layers for LLaMa. Overall, real-world pretrained Transformers have a much lower sparsity ratio than the assumptions.

**How does ICL evolve during training?** From the given constructions, models need to arrive at very specific weights to be able to perform gradient descent on in-context samples, but in practice, we observe models develop, retain, and improve this ability over time in training when the parameters change significantly (A detailed experimental setup is deferred to Appendix B). In Figure 2, we look at how the ability to perform ICL evolves compared with how the model parameters change over time (for each check-pointed GPT-J model). We measure the average parameter changes across all layers across $W_K, W_Q$, and $W_V$. This reveals that real Transformers do not settle on one set of weights (as required by previous works for performing GD) but continue to evolve throughout training. Although this result is an average over all the weights, certain groups of parameters (as constructed in previous works) are unlikely to remain constant throughout training. Therefore, ICL emerges in real LLMs, not just for a single choice of parameters but a family of parameters. Hence, **to prove the equivalence between GD and ICL, showing it for a single choice of parameters is not enough.**

## 4 ICL IS LIKELY NOT EQUIVALENT TO ORDER-STABLE ALGORITHMS

While we established some limiting assumptions in previous studies, it remains unclear whether ICL≈GD hypothesis is actually invalid for real LLMs (Ⓐ or Ⓑ in Figure 1). For two algorithms to be equivalent, they must also have the *same functional behavior*. Namely, they should respond identically to the changes in the ordering of the instances. In this section, we discuss the discrepant sensitivity of ICL and GD to the order in which they process training instances (demonstrations).

Let's begin with the definition of algorithmic equivalence.

**Definition 1** (Algorithmic equivalence to ICL). *Consider an optimization algorithm $\mathcal{A}$ that modifies a pretrained model $M_{\Theta_0} \in \mathcal{M}$, using demonstrations $S = \{(x_i, f(x_i)\}_{i=1}^N$ of a well defined task $f \sim \mathcal{F}$, i.e., $\Theta_S \leftarrow \mathcal{A}(S, M_{\Theta_0})$. We call $\mathcal{A}$ "equivalent" to ICL if and only if the following holds:*

$$M_{\Theta_0}(S_1 \circ S_2 \circ ... S_N \circ x^t) = M_{\Theta_S}(x^t) \quad \forall x_i, x^t \overset{f}{\sim} \mathcal{X}. \tag{2}$$

The following theorem establishes the equivalence of order sensitivity between ICL and any algorithm $\mathcal{A}$ equivalent to it:

**Theorem 1** (Algorithmic equivalence implies the same order sensitivity). *Given a pretrained model $M_{\Theta_0} \in \mathcal{M}$, an algorithm $\mathcal{A}$ equivalent to ICL, and demonstrations $S = \{(x_i, f(x_i))\}_{i=1}^N$ of a well defined task $f \sim \mathcal{F}$, let $\sigma_A, \sigma_B$ denote two orders of elements in S, such that $\Theta_{\sigma_A} \leftarrow \mathcal{A}(\sigma_A, M_{\Theta_0})$ and $\Theta_{\sigma_B} \leftarrow \mathcal{A}(\sigma_B, M_{\Theta_0})$. Then,*

$$\underbrace{M_{\Theta_0}(\sigma_A \circ x^t) - M_{\Theta_0}(\sigma_B \circ x^t)}_{\textit{The order sensitivity of ICL}} = \underbrace{M_{\Theta_{\sigma_A}}(x^t) - M_{\Theta_{\sigma_B}}(x^t)}_{\textit{The order sensitivity of algorithm } \mathcal{A}} \quad \forall x^t \stackrel{f}{\sim} \mathcal{X}. \tag{3}$$

*Proof.* The proof trivially follows from definition 1. We know that, $\forall x^t \stackrel{f}{\sim} \mathcal{X}$ we have:

$$M_{\Theta_0}(\sigma_A \circ x^t) = M_{\Theta_{\sigma_A}}(x^t), \qquad M_{\Theta_0}(\sigma_B \circ x^t) = M_{\Theta_{\sigma_B}}(x^t).$$

Simply subtracting these two terms proves the theorem. □

### 4.1 ICL IS LIKELY NOT GD BASED ON ORDER INCONSISTENCY

Let's assume that GD is equivalent to ICL (arrow Ⓐ in Figure 1). We show that this assumption leads to a contradiction due to their inconsistent order sensitivity.

**GD is order-stable.** We know that GD is performed on a batch of samples from the training distribution, as seen in Equation 1. It does not matter which order the samples are presented. GD calculates the gradient using the average loss across all samples and is therefore agnostic of the order in which they are calculated. With respect to theorem 1, if $\mathcal{A} = $ GD, $M_{\Theta_{\sigma_A}} = M_{\Theta_{\sigma_B}}$ or $M_{\Theta_{\sigma_A}}(x^t) - M_{\Theta_{\sigma_B}}(x^t) = 0$.

**ICL and GD show different order-sensitivity.** For ICL to be equivalent to any order-stable algorithm like GD, it must also be order-stable. However, previous research (Lu et al., 2022; Hahn & Goyal, 2023) has demonstrated that ICL is highly sensitive to the order of in-context samples. This is also easy to see because decoder-only Transformers exhibiting ICL only predict a token based on what they have seen before in the input. A different order of samples would change the behavior of the model. Therefore, ICL can not be equivalent to GD (arrow Ⓐ in Figure 1) as claimed by Dai et al. (2022). These conclusions may change upon notable technological shifts (e.g., the architecture of LLMs). We also empirically verify this phenomenon by comparing the output distributions produced by ICL and GD (Figure 4). (Details deferred to Appendix A.)

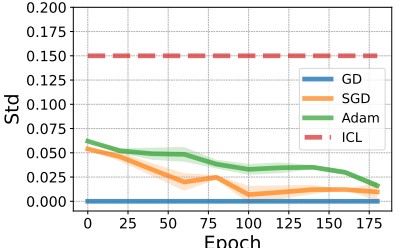

Figure 4: Order Sensitivity (standard deviation in output probabilities over the vocabulary) of ICL and GD (and its variants SGD and Adam) as measured on the LLaMa-7B on AGNews. The std is taken across 10 different orders of 8 ICL demos. More results are deferred to appendix A.

### 4.2 ICL IS LIKELY NOT $\widehat{\text{GD}}$ BASED ON ORDER INCONSISTENCY

**Gradient Descent on *implicit sub-model* ($\widehat{\text{GD}}$).** Akyürek et al. (2022); von Oswald et al. (2023) also hypothesize the existence of *implicit sub-models* inside the weights of Transformer models. These sub-models (parameterized to perform linear regression) are constructed into the weights of the Transformer. When the Transformer is presented with in-context samples, it can simulate steps of gradient descent on the regression loss (using these samples) with respect to the sub-model parameters. Formally, for a sub-model with weights $W_0$, the Transformer model $M_{\Theta_0} = M_{\Theta_0 \setminus W_0, W_0}$ with fixed parameters ($\Theta_0 \setminus W_0$) would optimize the weights of the inbuilt implicit sub-model ($W_0$) when presented with in-context samples and make its final prediction using updated weights ($W_1$). We refer to this version of GD as $\widehat{\text{GD}}$.

We first define the equivalence of ICL to an algorithm that updates the implicit model only.

**Definition 2.** *Consider an optimization algorithm $\mathcal{A}$ that modifies the implicit sub-model weights $W_0$ of a pretrained model $M_{\Theta_0} \in \mathcal{M}$, using demonstrations $S = \{(x_i, f(x_i))\}_{i=1}^N$ of a well defined task $f \sim \mathcal{F}$, i.e., $W_S \leftarrow \mathcal{A}(S, W_0)$. We call $\mathcal{A}$ "equivalent" to ICL if and only if the following holds:*

$$M_{\Theta_0 \setminus W_0, W_0}(S_1 \circ S_2 \circ ... S_N \circ x^t) = M_{\Theta_S \setminus W_S, W_S}(x^t) \quad \forall x_i, x^t \stackrel{f}{\sim} \mathcal{X}, \tag{4}$$

*and $\Theta_0 \setminus W_0 = \Theta_S \setminus W_S$, i.e., the pretrained model only updates by the sub-models weights.*

When the model with implicit sub-model weights $W_0$ is provided with in-context examples, it arrives at updated weights $W_S$ using $\mathcal{A}$ without changing any other weights. This is equivalent to when the model starts with sub-model weights $W_S$ and is provided no in-context examples, so no update happens on the weights via $\mathcal{A}$. Now, based on Definition 2 and Theorem 1, the following corollary about the equivalence of order sensitivity between ICL and an equivalent algorithm $\mathcal{A}$ also holds:

**Corollary 1.** *For a pretrained model $M_{\Theta_0} \in \mathcal{M}$, an algorithm $\mathcal{A}$ equivalent to ICL (according to definition 2) and two orders $\sigma_A, \sigma_B$ of elements in the demonstration set $S$, $\forall\, x^t \overset{f}{\sim} \mathcal{X}$,*

$$M_{\Theta_0 \setminus W_0, W_0}(\sigma_A \circ x^t) - M_{\Theta_0 \setminus W_0, W_0}(\sigma_B \circ x^t) = M_{\Theta_{\sigma_A} \setminus W_{\sigma_A}, W_{\sigma_A}}(x^t) - M_{\Theta_{\sigma_B} \setminus W_{\sigma_B}, W_{\sigma_B}}(x^t) \quad (5)$$

**ICL and $\widehat{\text{GD}}$ show different order-sensitivity.** Let's assume that $\widehat{\text{GD}}$ is equivalent to ICL (arrow Ⓑ in Figure 1) according to definition 2. According to the same argument as in §4.1, $W_{\sigma_A} = W_{\sigma_B}$ or $\Theta_{\sigma_A} \setminus W_{\sigma_A}, W_{\sigma_A} = \Theta_{\sigma_B} \setminus W_{\sigma_B}, W_{\sigma_B}$ or $M_{\Theta_{\sigma_A} \setminus W_{\sigma_A}, W_{\sigma_A}}(x^t) - M_{\Theta_{\sigma_B} \setminus W_{\sigma_B}, W_{\sigma_B}}(x^t) = 0$. This again implies that for ICL to be equivalent to $\widehat{\text{GD}}$, it must be order-stable. As shown previously, empirical evidence shows that ICL is not order-stable and hence not equivalent to $\widehat{\text{GD}}$ (arrow Ⓑ in Figure 1), assuming today's LLM technology. These conclusions may change in future.

**What about variants of GD?** We note that the construction of Akyürek et al. (2022) allows for order sensitivity in GD as the update is performed on samples one by one instead of the batch update performed by von Oswald et al. (2023). Although it is unclear which order is used to perform this update, we compared the order-sensitivity of ICL with SGD and Adam (Figure 4) and found that ICL is still significantly more sensitive to order than SGD/Adam. Therefore, we believe it is unlikely that ICL is equivalent to even variants of GD. We provide more order-sensitivity results in Appendix A.

# 5 EMPIRICAL EVALUTATION OF ICL VS. GD/$\widehat{\text{GD}}$ IN LLMS

This section provides an empirical evaluation of ICL≈GD equivalence in realistic settings. Specifically, we take a language model pretrained on natural data and use it with ICL demos to get ICL outputs. Then, we use the same demos to fine-tune the model using GD and $\widehat{\text{GD}}$, and get their respective output (without ICL demos). Next, we compare these outputs on various metrics to see how well ICL and GD/$\widehat{\text{GD}}$ align in practice.

## 5.1 EXPERIMENTAL SETTINGS

**Model and benchmarks.** We choose LLaMa (7B) (Touvron et al., 2023) as our primary model for evaluation. Our model-size comparative studies use the GPT family of models (as discussed later §5.2). For benchmarking, we select the following datasets: AGNews (Zhang et al., 2015), CB (De Marneffe et al., 2019), SST-2 (Socher et al., 2013), RTE (Dagan et al., 2005). In the main text, we show results on AGNews and defer other corresponding results to Appendix E.

**Experimental setup.** We evaluate ICL with varying demonstration sizes $N \in \{1, 2, 4, 8\}$ and for GD, we fine-tune the models with the same corresponding ICL demonstrations, experimenting with a variety of learning rates $\{$1e-4, 5e-4, 1e-5, 5e-5$\}$ over 200 epochs, which ensures the convergence of model. Specifically, the objective function of GD is $\mathcal{J} = \sum_{(x,y) \in S} \mathcal{L}_{\text{clm}}(y; x)$, where $\mathcal{L}_{\text{clm}}(y; x)$ is the CLM loss of $y$, given $x$ as the prefix. For $\widehat{\text{GD}}$, it is not trivial to identify the *implicit* sub-model as described in §4.2. Moreover, it is computationally infeasible to experiment on all possible subsets of parameters to identify the sub-model. Therefore, we use the hypotheses in (Akyürek et al., 2022; von Oswald et al., 2023), to experiment with intuitive subsets. In particular, according to von Oswald et al. (2023) the *implicit* model lies in $W_V$ of the Transformer while the probing experiments in Akyürek et al. (2022) suggest that this iterative optimization happens in top layers of the Transformers. Therefore, we provide experiments with three intuitive subsets to simulate $\widehat{\text{GD}}$: finetuning (1) all weights of multiple layers while keeping others fixed, (2) $W_V$ of a single deep layer, and (3) $W_V$ of a single intermediate layer. Details are deferred to Appendix F.

**Evaluation metrics.** Here are the metrics used for our analysis (further details in Appendix D).

*Performance*: We compute the accuracy against the token with the highest mass from the whole vocabulary $V$ (rather than the label set $\mathcal{Y}$).

*Token Overlap*: This is a relative metric comparing two distributions over vocabulary $V$. We sort the tokens based on their probability mass for each distribution and select the top-$K$ tokens (denoted by $T_K^1$ and $T_K^2$). The token overlap is calculated as $\frac{1}{K}|T_K^1 \cap T_K^2|$. We use $K = 10$ in our experiments.

*Overlap Cosine Similarity* (OCS): Similar to *Token Overlap*, this is a relative metric between two distributions $p^1$, $p^2$ over vocabulary $V$. However, this metric differs because it accounts for confidences assigned to each token. Specifcially. It is defined as follows: $\frac{\sum_{t \in O} p^1(t) \cdot p^2(t)}{\sqrt{(\sum_{t \in O} p^1(t)^2) \cdot (\sum_{t \in O} p^2(t)^2) \cdot (K - |O|)}}$, where $O = T_K^1 \cap T_K^2$. Intuitively, this quantifies the cosine distance between the overlapping tokens and assumes all the other tokens have zero overlap, therefore normalizing by $\sqrt{(K - |O|)}$ (when $K = |O|$, we divide by $\sqrt{1}$).

We evaluate each metric across three random seeds and compute the average and variance. Each random seed is used to sample demos and their order for use in ICL experiments. Each seed uses the same demos in ICL and GD/$\widehat{GD}$ for consistency.

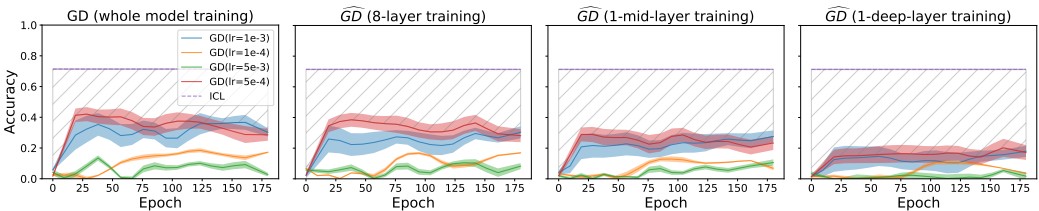

(a) $y$-axis shows 'performance' comparison of ICL with GD variants.

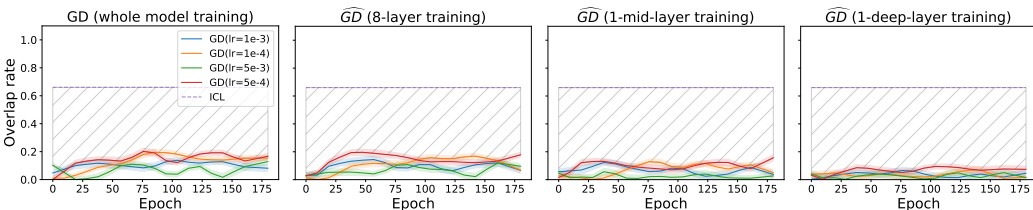

(b) $y$-axis shows 'Token Overlap' of ICL in comparison with GD variants.

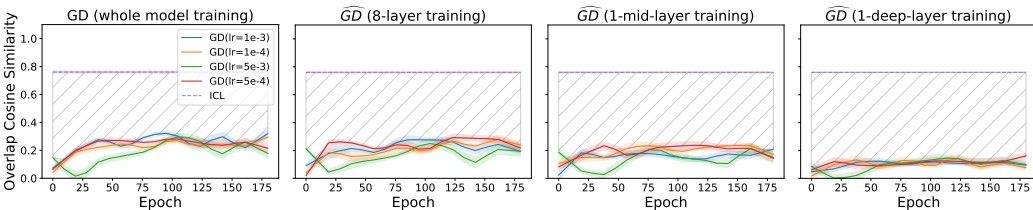

(c) $y$-axis shows 'Overlap Cosine Similarity' of ICL in comparison with GD variants.

Figure 5: Comparison of ICL and GD/$\widehat{GD}$ on our three metrics for the AGNews dataset (results with 4 ICL demos). Note that the ICL plots for each metric is substantially different from the corresponding GD plot, showing the substantial gap between ICL and GD (highlighted by the gray diagonal lines).

## 5.2 RESULTS

With the experimental setup defined in §5.1, we investigate the gap between ICL and GD/$\widehat{GD}$ with our three evaluation metrics. The results, as seen in Figure 5, highlight a notably consistent gap between ICL and GD/$\widehat{GD}$ across all three metrics. For some set of experimental parameters, the performance metrics might be similar (as also shown in Dai et al. (2022)), but the other nuanced metrics (token and confidence overlap) between the two are significantly different, which must result

from a different functional dynamic. This is true for both GD and $\widehat{\text{GD}}$, implying that the dynamics of fine-tuning any or all sets of weights on in-context samples results differs greatly from the mysterious ICL. Results on other datasets, and with different numbers of ICL demos are similar and are deferred to Appendix E (GD) and Appendix F ($\widehat{\text{GD}}$). More results about impact of model size can be seen in Appendix G.

## 6 RELATED WORK

We review the relevant literature on the functional interpretation of in-context learning via GD. We delegate other explanations of ICL to Appendix H as the are tangential to our focus.

Many works offer functional explanations of ICL (Liu et al., 2022; Olsson et al., 2022; Schlag et al., 2021). Among these, explanations via GD Garg et al. (2022); Zhang et al. (2023); Ahn et al. (2023) are most pertinent to our work. Notably, Akyürek et al. (2022) showed that Transformers can implement learning algorithms (gradient descent or closed-form OLS) for linear regression problems and empirically showed that the optimality of algorithms implemented experience a *phase shift* with increasing model size. Raventós et al. (2023) discover similar results about algorithm discovery and phase shifts with increasing task diversity. Dai et al. (2022) similarly show a dual between attention layers and linear layers optimized using gradient descent. Li et al. (2023) show such an equivalence on softmax regression tasks. Finally, von Oswald et al. (2023) showed a similar construction with a simpler Linear Self-Attention Transformer, claiming that Transformers learn in-context using gradient descent on linear regression problems. Notably, Akyürek et al. (2022) found this GD behavior applicable only in small models, with bigger models exhibiting Bayes optimal learning behavior (like Ordinary Least Squares for linear regression). In contrast, von Oswald et al. (2023) claim that bigger Transformers also implement GD with added data transformations.

Most of this line of work shows how Transformers have the ability to implement such algorithms resulting from training on ICL objectives (**hypothesis 2**) and not that real-world models pretrained on natural data develop this ability (**hypothesis 1**).

## 7 DISCUSSION AND CONCLUSION

This work intends to clarify the distinction between naturally emergent ICL (commonly seen in LLMs pretrained on natural text data); hypothesis 1) vs. task-specific ICL as a result of training Transformers for ICL (hypothesis 2). While recent work has shown that Transformers have the ***expressive capacity*** to simulate gradient-descent in their forward pass, this does ***not*** immediately imply that real-world models ***actually do*** simulate it. We hope this work motivates alternative approaches that reveal the true nature of in-context learning in pretrained LLMs.

We recognize that hypothesis 1 establishing a universal equivalence between ICL and GD may be too strong. A more reasonable hypothesis might involve certain restrictions, such as the target task's distributional properties or the number of demonstrations. However, the specifics of such conditions are unclear, so we have opted for a general statement.

Besides using in-context demonstrations, recent work has also discovered other ways in which in-context prompts enhance the performance of LLMs. For example, appending prompts like "*Think step by step*" (Kojima et al., 2022) or "*Take a deep breath and think*" (Yang et al., 2023) before asking a task-specific question has been shown to improve zero-shot performance of LLMs. Such evidence may suggest that an optimization algorithm like GD cannot fully describe the ability of ICL. Understanding ICL dynamics requires a more holistic theory, considering the various nuances of this remarkable learning paradigm.

### LIMITATIONS AND FUTURE OPPORTUNITIES

Because of its computationally infeasible nature, we were not able to do an exhaustive search over all sub-models and pinpoint which subset of parameters could correspond to sub-models that could get updated in $\widehat{\text{GD}}$. This could be an interesting avenue of research. Moreover, we do not provide alternate explanations of how ICL works functionally, which we aim to do in the future.

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

# SUPPLEMENTARY MATERIAL

## A    ORDER SENSITIVITY OF ICL AND GD-BASED ALGORITHMS (EXPERIMENTAL SETUP)

We present empirical evidence highlighting the distinct sensitivities of GD-based algorithms and ICL with respect to data order. Specifically, we assess the variation in confidence assigned to vocabulary $V$ by the model across different data orderings.

**Experimental setup**    We evaluate the order sensitivity of GD-based algorithms using the GD, SGD, and Adam optimizers. The chosen learning rates are 1e-4, 1e-5, 5e-4, and 5e-5. Our experiments are conducted on the AGNews dataset using the LLaMa-7B model. We set the number of demonstrations to 8, and for SGD and Adam, the mini-batch size is fixed at 2. GD training continues for 200 epochs to guarantee convergence. The number $N$ of orders $\{\sigma_i\}_{i=1}^{N}$ is set as 10.

**Evaluation metric (Sen)**    As for the evaluation metric of sensitivity (Sen), it is defined as follows: Given a set of confidence vectors $\{p_i\}_{i=1}^{N}$ resulting from distinct data orders $\{\sigma_i\}_{i=1}^{N}$, we calculate the standard deviation for each dimensionality within $V$ using the samples $\{p_i\}_{i=1}^{N}$. Subsequently, the variances for individual tokens are aggregated.

**Results**    In Figure 4, we present the results highlighting several key observations. First, ICL exhibits a much more pronounced data order sensitivity than the three GD-based algorithms. Second, as GD training progresses, its sensitivity diminishes, widening the divergence towards ICL. Overall, these findings underscore distinct behaviors of ICL and GD-based algorithms with respect to data order. This suggests a disparity between ICL and GD, as shown in Theorem 1.

**Additional results on order sensitivity of ICL and GD-based algorithm**    In this part, we provide extra evidence showing the order sensitivity of ICL and GD. Specifically, we consider both GD and $\widehat{\text{GD}}$ (the same implementation as in Appendix F) and then vary the batch size of each version. Following the experimental setup in §4, we show the results in Figure 6 and Figure 7. We can observe that both GD and $\widehat{\text{GD}}$ have substantially different order sensitivity towards ICL.

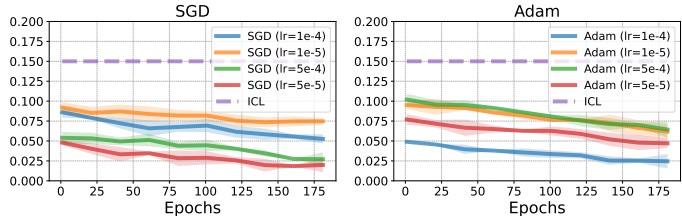

(a) Order sensitivity of ICL and GD when batchsize = 1

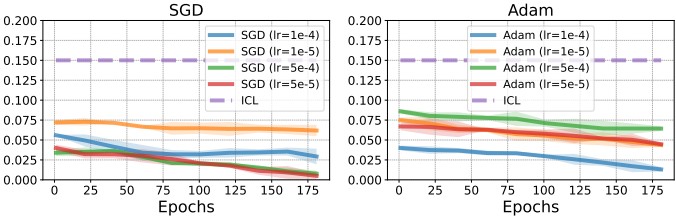

(b) Order sensitivity of ICL and $\widehat{\text{GD}}$ when batchsize = 4

Figure 6: The order sensitivity (y-axis represents Sen (appendix A)) of ICL and GD (SGD and Adam) as the batchsize changes.

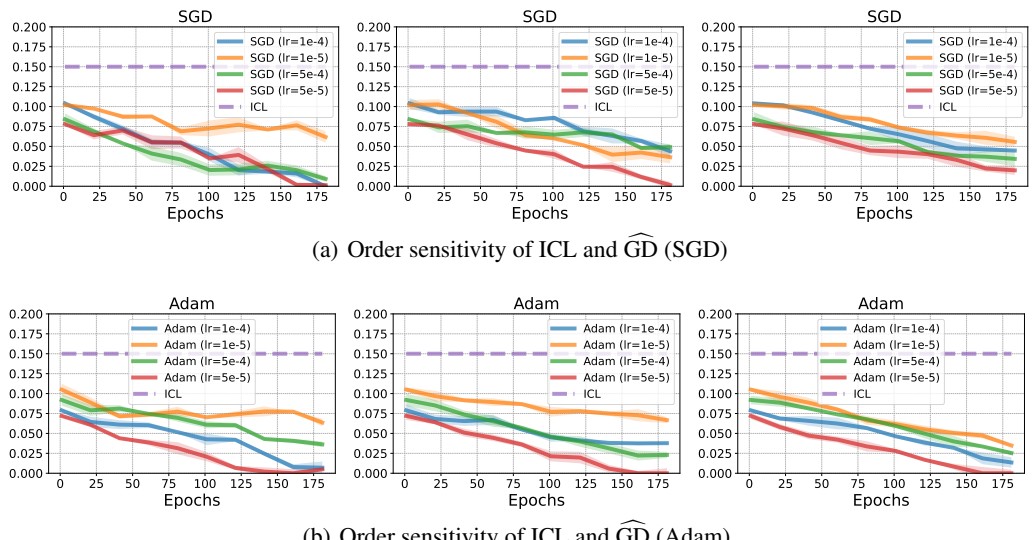

(a) Order sensitivity of ICL and $\widehat{\text{GD}}$ (SGD)

(b) Order sensitivity of ICL and $\widehat{\text{GD}}$ (Adam)

Figure 7: The order sensitivity (y-axis represents Sen (appendix A)) of ICL and $\widehat{\text{GD}}$ (SGD and Adam) as the batchsize changes. From left to right, three figures refer to cases bs=1, 2, 4.

## B  HOW DOES ICL EVOLVE DURING TRAINING (EXPERIMENTAL SETUP)

**Experimental setup.**  We chose intermediate checkpoints from GPT-J, ranging from 310k to 380k pretraining steps. Using these varied pretraining steps, our approach simulates the fine-tuning process. Specifically, we focus on two metrics to quantify the magnitude of fine-tuning: (1) Step Gap: This represents the difference in pretraining steps between selected checkpoints. (2) Parameter Gap: In line with the assumptions made by Oswald et al. (von Oswald et al., 2023), we compute the average differences for each parameter within the $W_K$, $W_Q$, and $W_V$ matrices across different checkpoints. To evaluate the ICL capacity of the models, we conducted tests on AGNews, SST-2, CB, and RTE using eight demonstrations.

**Results.**  The results are shown in Figure 8, from where we can observe that there is no significant gap between ICL capacity of different checkpoints, indicating that continued fine-tuning (pretraining) will not substantially hurt the ICL performance.

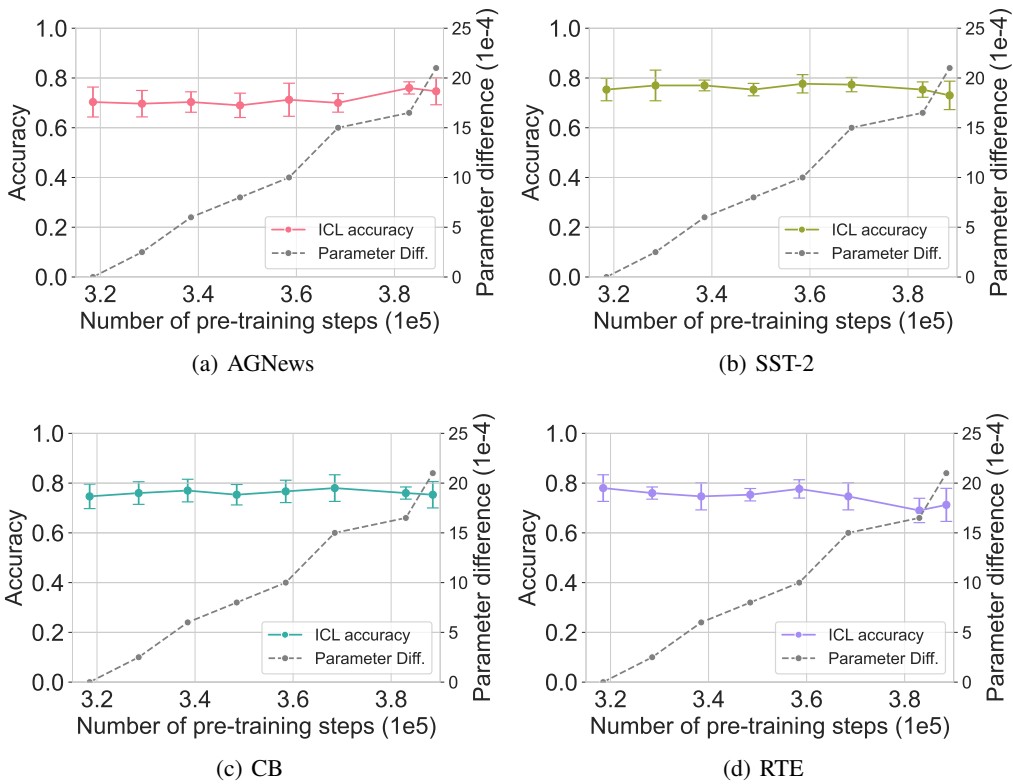

Figure 8: The ability of GPT-J to perform ICL does not change much over a time cross-section of training while the parameters change steadily.

## C LAYER-WISE SPARSITY RATE OF LLMS

We show the sparsity ratio of each layer of LLMs. Specifically, in our paper, we have used LLaMa-7B and GPT-J are main experiments, so we show their sparsity rate of $W_K$, $W_Q$, and $W_V$ in each layer. The results are shown in Figure 9. It is interesting that although $W_K$ and $W_Q$ have almost constant sparsity in all layers, $W_V$ has slightly decaying sparsity.

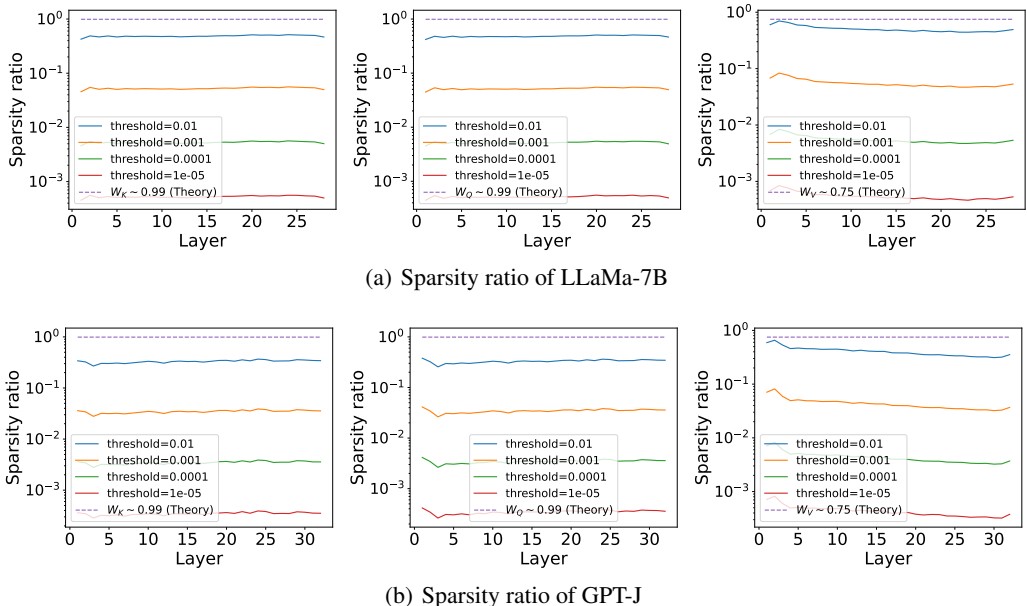

(a) Sparsity ratio of LLaMa-7B

(b) Sparsity ratio of GPT-J

Figure 9: The sparse ratio of LLaMa-7B and GPT-J in each layer. From left to right, three figures represent the cases of $W_K$, $W_Q$, and $W_V$.

# D  EVALUATION METRICS IN OUR EXPERIMENTS

Previous works often use performance metrics (accuracy and loss) based on the token with the maximum probability from *label set* $\mathcal{Y}$, when comparing different algorithms (Srivastava et al., 2023; Wei et al., 2021). We argue that these metrics do not paint the whole picture. Even if two sorting algorithms reach the same result, their dynamics may differ. In particular, we argue that the **relative uplifting** of tokens in the output distribution sheds light on the dynamics of the algorithm. For this purpose, we propose two more metrics that help evaluate the equivalence between ICL and GD dynamics and change the matching criteria.

- *Performance*: As all of our datasets have classification tasks, we compute the accuracy by picking the top token from the whole vocabulary $V$ instead of just the label set $\mathcal{Y}$. Checking whether the correct token made it to the top evaluates the precision of the optimization algorithm. It is defined as $\frac{1}{|S_{\text{test}}|} \sum_{(x_i^t, y_i^t) \in S_{\text{test}}} \mathbf{1}\{y_i^t = \arg\max M(C \circ x_i^t)\}$, where $M$ is the model, $C$ is the context and $S_{\text{test}}$ is the test set. Notice that this is an algorithm-specific metric, i.e., we can calculate it using a single output distribution produced by the model $M$.

- *Token Overlap*: This is a relative metric computed based on two output distributions. These distributions could be either produced by the same model on different inputs (in case of ICL: different number of demos, order of demos, etc.) or different models on the same inputs (ICL (with context) vs GD (fine-tuned, without context)). We compute the top-$K$ tokens (denoted by $T_K^1$ and $T_K^2$) from each output distribution and find the overlap between these sets. This illustrates how differently the two models (or inputs) change the probability weights of tokens, evaluating the difference between their dynamics. The token overlap metric is calculated as $\frac{1}{K}|T_K^1 \cap T_K^2|$. We use $K = 10$ as it fairly represents most of the probability mass of the output distribution.

- *Overlap Cosine Similarity* (OCS): Token overlap evaluates each of the top-$K$ tokens with the same weight. With confidence overlap, we measure the agreement at a finer level, looking at how well the tokens agree individually. This metric is computed on the confidence distribution on top-$K$ tokens. This is done because the vocabulary set is large, and most tokens have low probabilities, so the overall cosine similarity is always close to $1$. As the top-$K$ tokens may not be the same in the two distributions being compared, we denote the intersection of the two sets $T_K^1, T_K^2$ by $O = T_K^1 \cap T_K^2$ and use the following formula: $\frac{\sum_{t_i \in O} p^1(t_i) \cdot p^2(t_i)}{\sqrt{(\sum_{t_i \in O} p^1(t_i)^2) \cdot (\sum_{t_i \in O} p^2(t_i)^2) \cdot (K - |O|)}}$. It measures the cosine distance between the overlapping tokens and assumes all the other tokens have zero overlap, therefore normalizing by $\sqrt{(K - |O|)}$ (when $K = |O|$, we divide by $\sqrt{1}$).

We evaluate every metric across three random seeds and compute the average and variance. Each random seed is used to sample demos for use in ICL experiments. The same demos are used to fine-tune models for GD. For the relative metrics (*Token Overlap* and *Confidence Overlap*), the values for ICL are calculated between predictions made for the same set of demos but in a different order, also underlining the high-order sensitivity of ICL.

# E  ADDITIONAL RESULTS ON ICL VS GD COMPARISONS

Here are the extra results on ICL vs GD on other benchmarks beyond AGNews in §5.

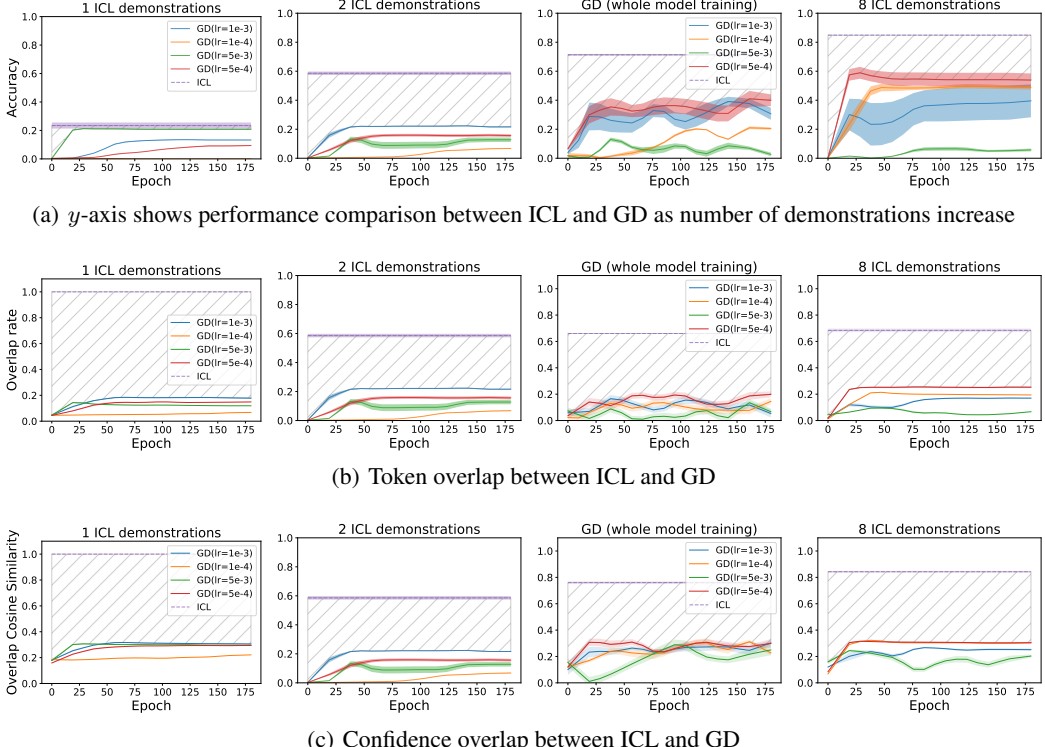

(a) $y$-axis shows performance comparison between ICL and GD as number of demonstrations increase

(b) Token overlap between ICL and GD

(c) Confidence overlap between ICL and GD

Figure 10: Comparison of ICL and GD on our three evaluation metrics for the AGNews dataset.

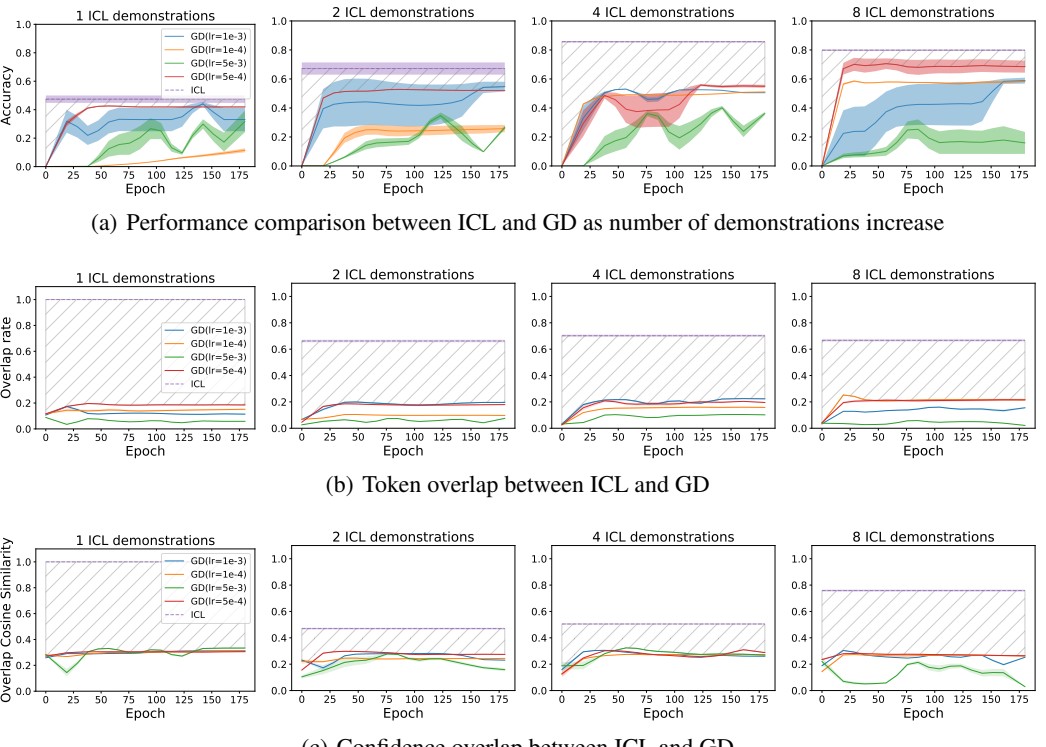

(a) Performance comparison between ICL and GD as number of demonstrations increase

(b) Token overlap between ICL and GD

(c) Confidence overlap between ICL and GD

Figure 11: Comparison of ICL and GD on our three evaluation metrics for the SST dataset.

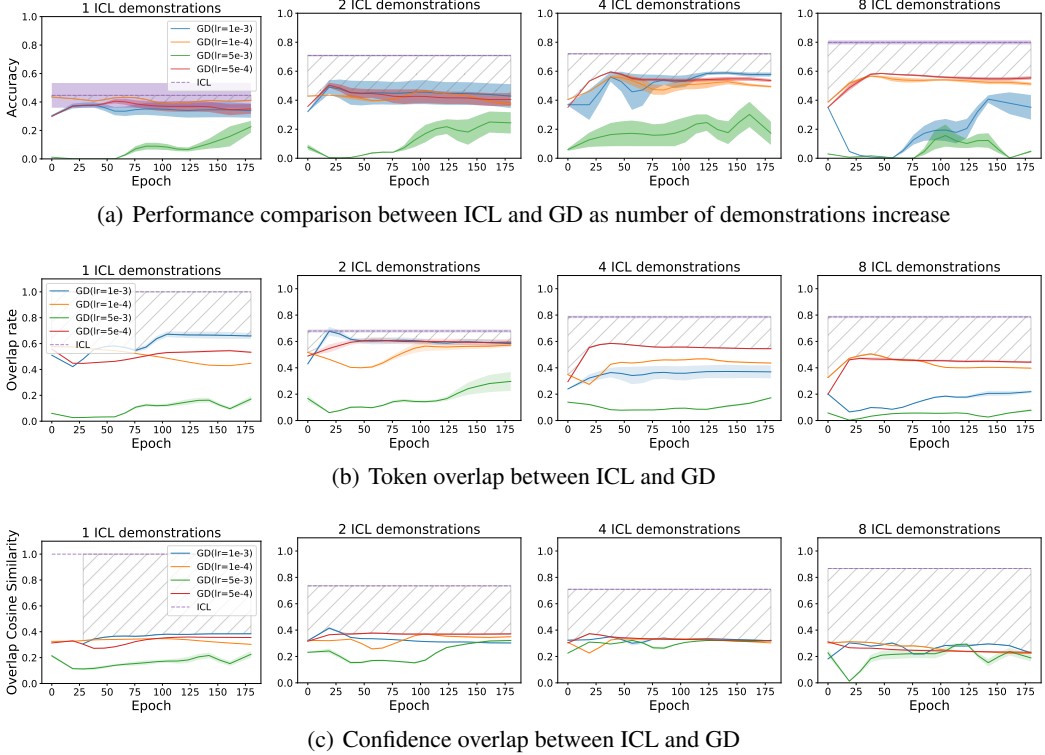

(a) Performance comparison between ICL and GD as number of demonstrations increase

(b) Token overlap between ICL and GD

(c) Confidence overlap between ICL and GD

Figure 12: Comparison of ICL and GD on our three evaluation metrics for the CB dataset.

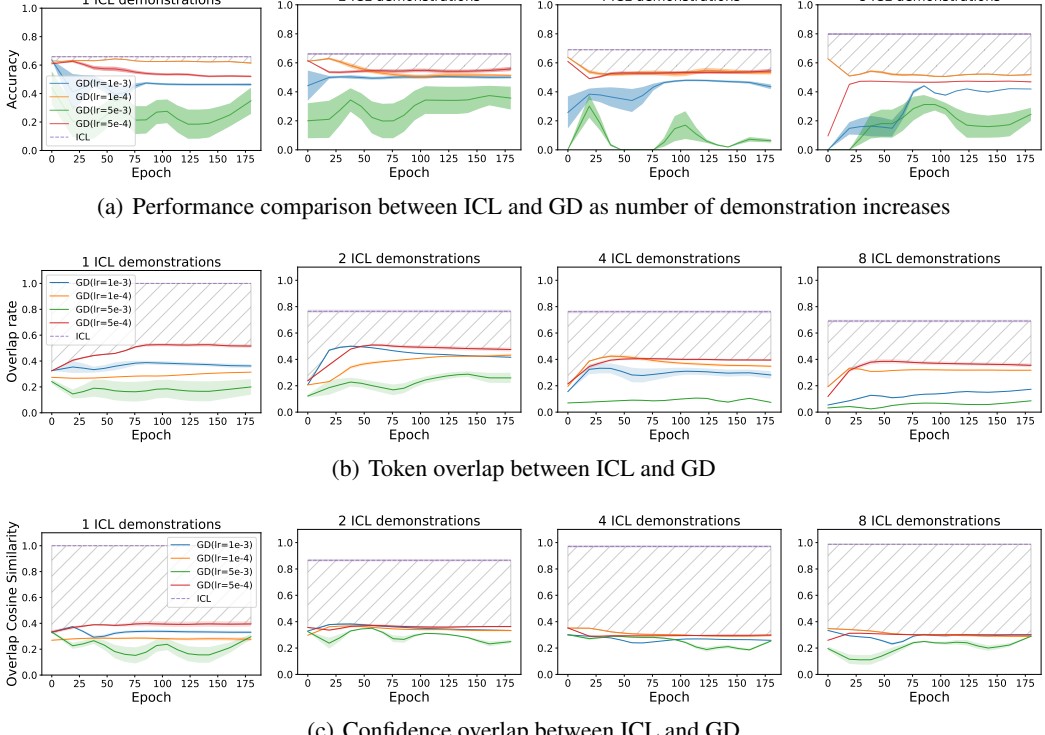

(a) Performance comparison between ICL and GD as number of demonstration increases

(b) Token overlap between ICL and GD

(c) Confidence overlap between ICL and GD

Figure 13: Comparison of ICL and GD on our three evaluation metrics for the RTE dataset.

# F   EMPIRICAL RESULTS ON ICL VS $\widehat{\text{GD}}$

In this section, we present empirical results on ICL vs $\widehat{\text{GD}}$.

**How are sub-models selected for optimization?**   Since $\widehat{\text{GD}}$ conducts updates only on the subset of the model and enumerating all the possible subsets of model parameters is infeasible, we select intuitive subsets of parameters to simulate $\widehat{\text{GD}}$.

We use the hypotheses in (Akyürek et al., 2022; von Oswald et al., 2023), to experiment with intuitive subsets. In particular, according to von Oswald et al. (2023) the *implicit* model lies in $W_V$ of the Transformer while the probing experiments in Akyürek et al. (2022) suggest that this iterative optimization happens in top layers of the Transformers. Therefore, we provide experiments with three intuitive subsets to simulate $\widehat{\text{GD}}$: finetuning (1) all weights of multiple layers while keeping others fixed, (2) $W_V$ of a single deep layer, and (3) $W_V$ of a single intermediate layer.

**Results of ICL vs. $\widehat{\text{GD}}$ (Deep layers)**   Following a similar experimental setup in §5, we compare the differences between ICL and $\widehat{\text{GD}}$. Here are the results for randomly selecting one layer from the last four layers from LLaMa (29-32); we repeat the experiments four times and plot the mean and std. The results are shown in Figure 14 - Figure 17, and we can observe the gaps between ICL and $\widehat{\text{GD}}$.

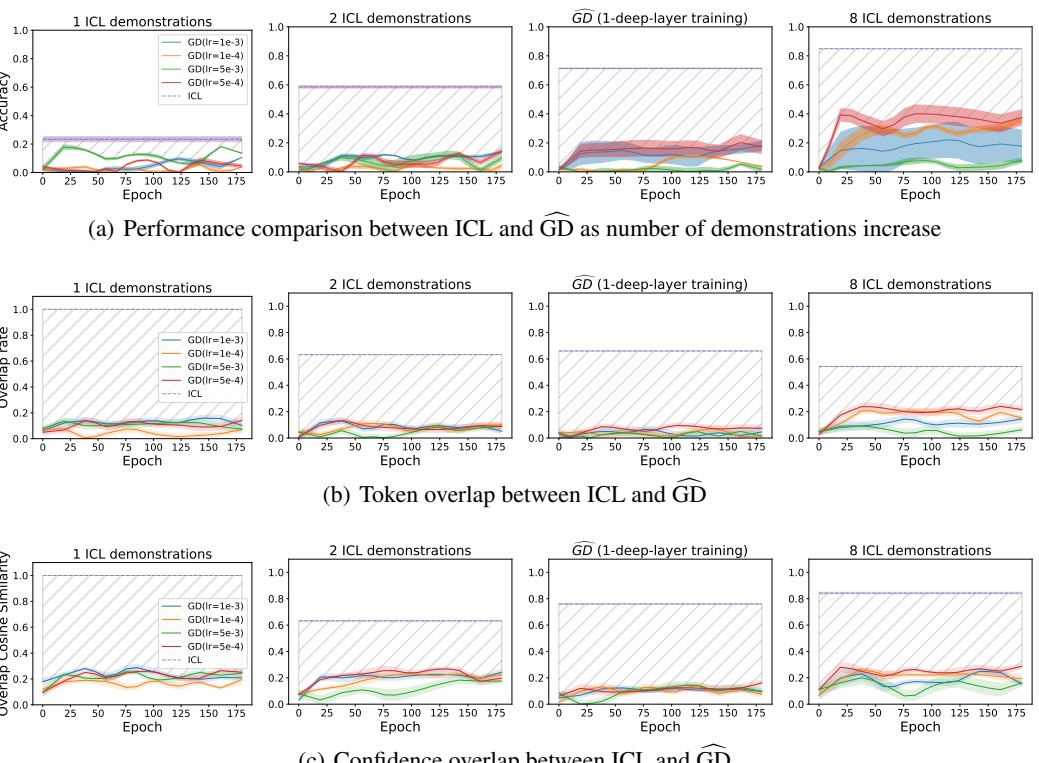

(a) Performance comparison between ICL and $\widehat{\text{GD}}$ as number of demonstrations increase

(b) Token overlap between ICL and $\widehat{\text{GD}}$

(c) Confidence overlap between ICL and $\widehat{\text{GD}}$

Figure 14: Comparison of ICL and $\widehat{\text{GD}}$ on our three evaluation metrics for the AGNews dataset. $\widehat{\text{GD}}$ is simulated by optimizing on one random deep layer of LLaMa.

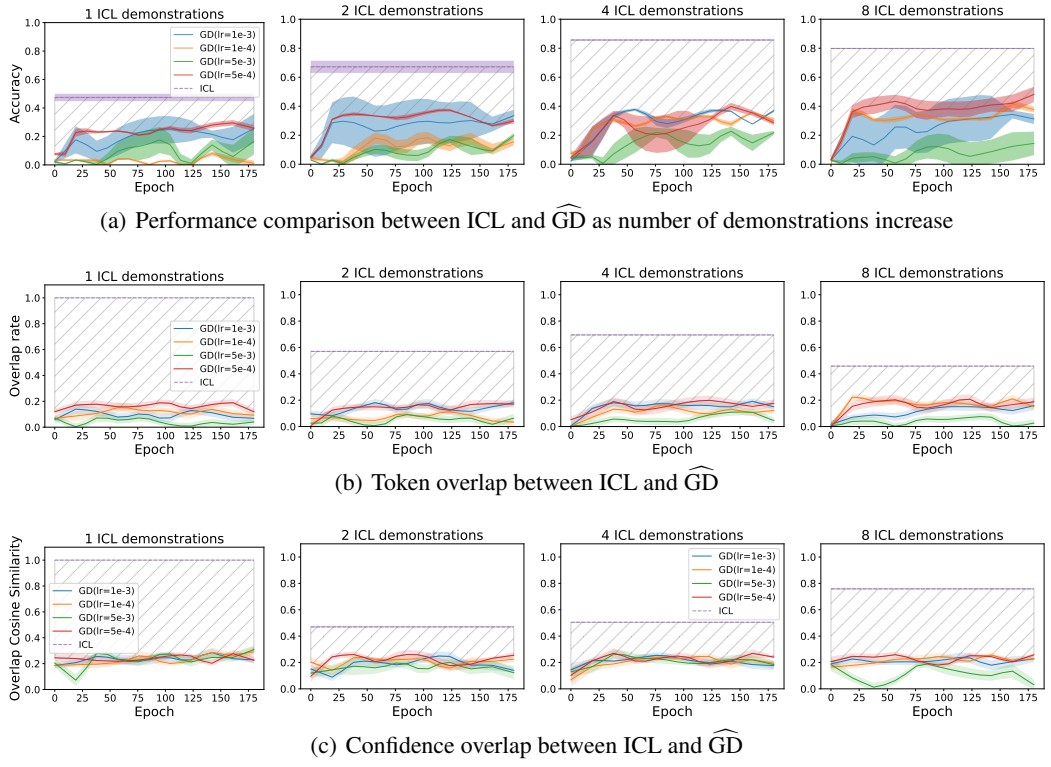

(a) Performance comparison between ICL and $\widehat{\text{GD}}$ as number of demonstrations increase

(b) Token overlap between ICL and $\widehat{\text{GD}}$

(c) Confidence overlap between ICL and $\widehat{\text{GD}}$

Figure 15: Comparison of ICL and $\widehat{\text{GD}}$ on our three evaluation metrics for the SST dataset. $\widehat{\text{GD}}$ is simulated by optimizing on one random deep layer of LLaMa.

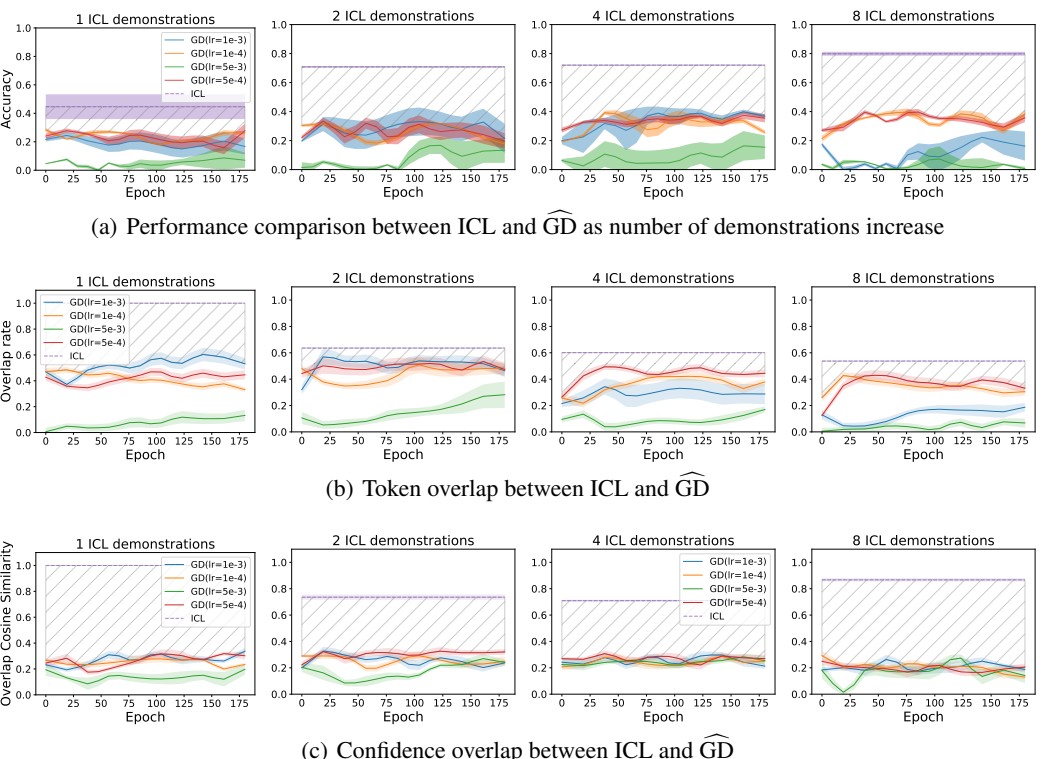

(a) Performance comparison between ICL and $\widehat{\text{GD}}$ as number of demonstrations increase

(b) Token overlap between ICL and $\widehat{\text{GD}}$

(c) Confidence overlap between ICL and $\widehat{\text{GD}}$

Figure 16: Comparison of ICL and $\widehat{\text{GD}}$ on our three evaluation metrics for the CB dataset. $\widehat{\text{GD}}$ is simulated by optimizing on one random deep layer of LLaMa.

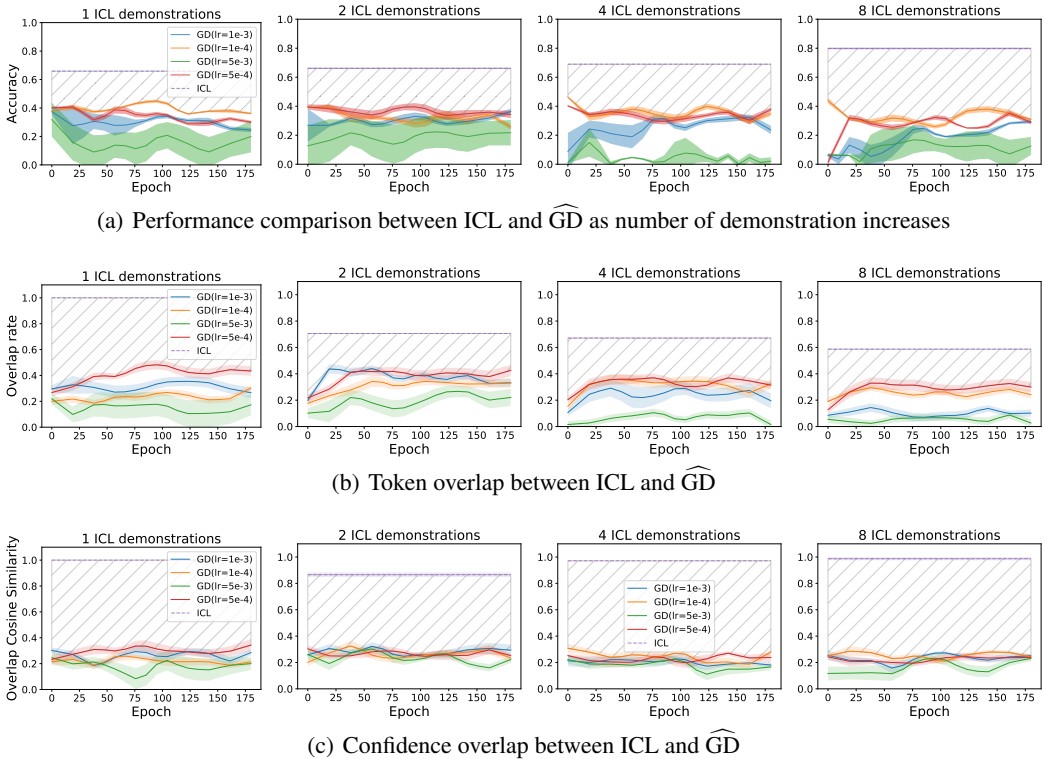

(a) Performance comparison between ICL and $\widehat{\text{GD}}$ as number of demonstration increases

(b) Token overlap between ICL and $\widehat{\text{GD}}$

(c) Confidence overlap between ICL and $\widehat{\text{GD}}$

Figure 17: Comparison of ICL and $\widehat{\text{GD}}$ on our three evaluation metrics for the RTE dataset. $\widehat{\text{GD}}$ is simulated by optimizing on one random deep layer of LLaMa.

**Results of ICL vs. $\widehat{\text{GD}}$ (Middle layers)** Following a similar experimental setup in §5, we compare the differences between ICL and $\widehat{\text{GD}}$. We randomly select one layer from the middle layers of LLaMa (16-20). Here are the results for randomly selecting one layer from LLaMa; we repeat the experiments four times and plot the mean and std. The results are shown in Figure 18 - Figure 21, we can observe the gaps between ICL and $\widehat{\text{GD}}$.

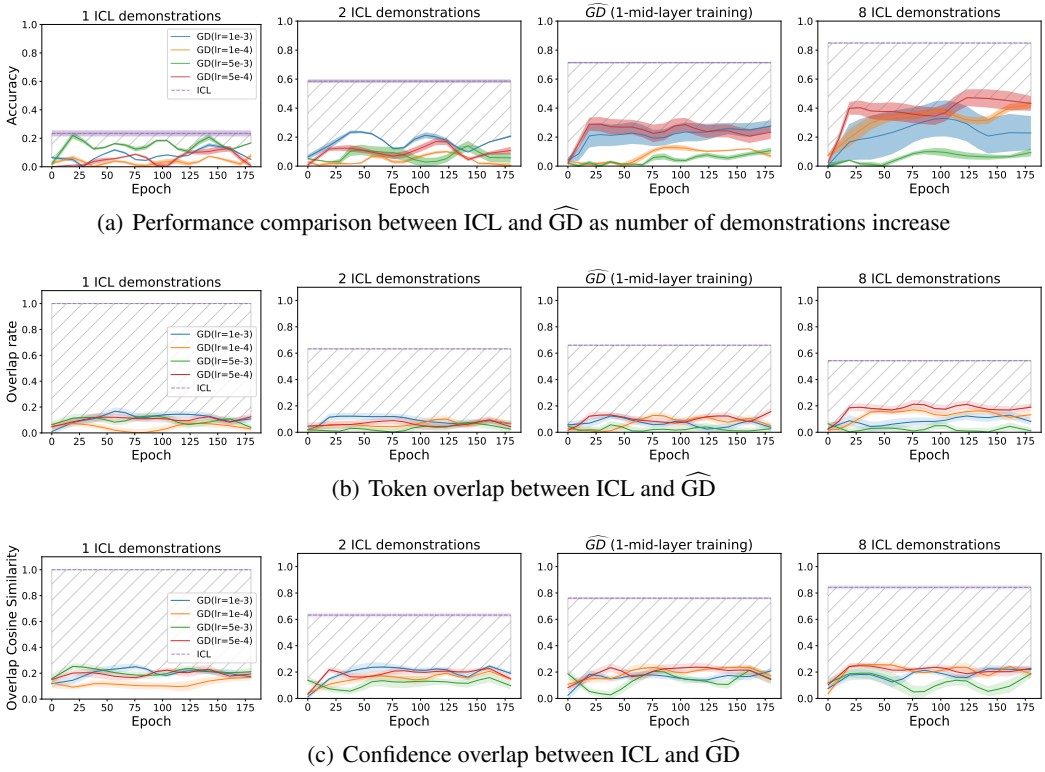

(a) Performance comparison between ICL and $\widehat{\text{GD}}$ as number of demonstrations increase

(b) Token overlap between ICL and $\widehat{\text{GD}}$

(c) Confidence overlap between ICL and $\widehat{\text{GD}}$

Figure 18: Comparison of ICL and $\widehat{\text{GD}}$ on our three evaluation metrics for the AGNews dataset. $\widehat{\text{GD}}$ is simulated by optimizing on one random middle layer of LLaMa.

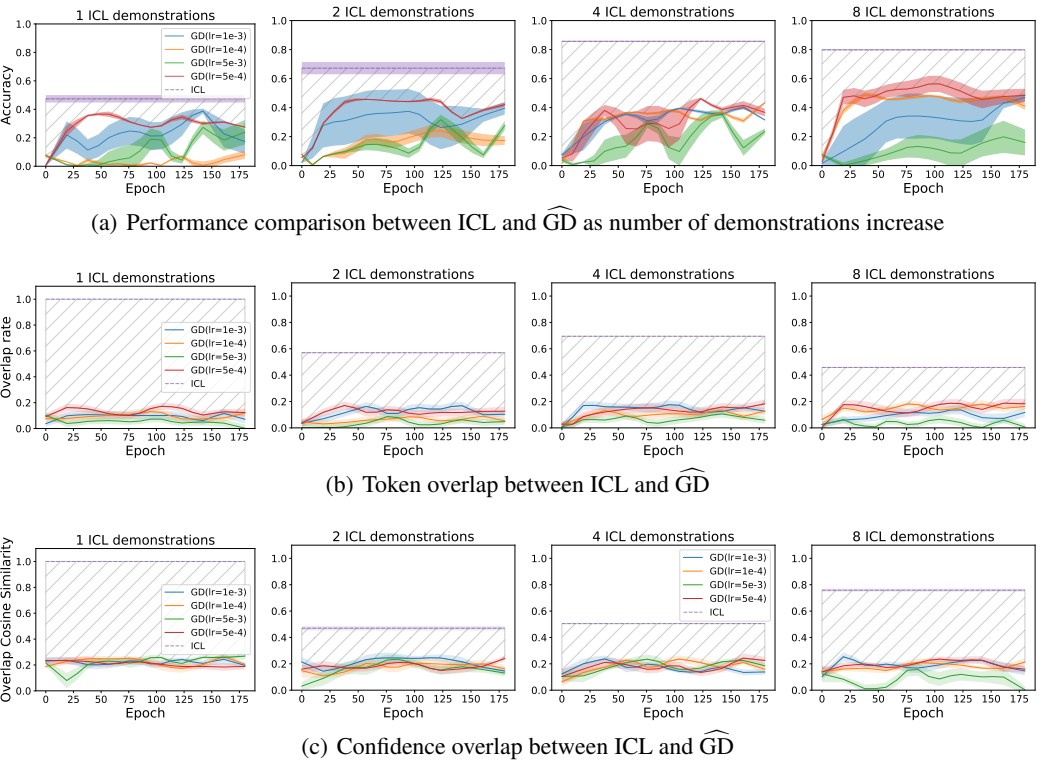

(a) Performance comparison between ICL and $\widehat{\text{GD}}$ as number of demonstrations increase

(b) Token overlap between ICL and $\widehat{\text{GD}}$

(c) Confidence overlap between ICL and $\widehat{\text{GD}}$

Figure 19: Comparison of ICL and $\widehat{\text{GD}}$ on our three evaluation metrics for the SST dataset. $\widehat{\text{GD}}$ is simulated by optimizing on one random middle layer of LLaMa.

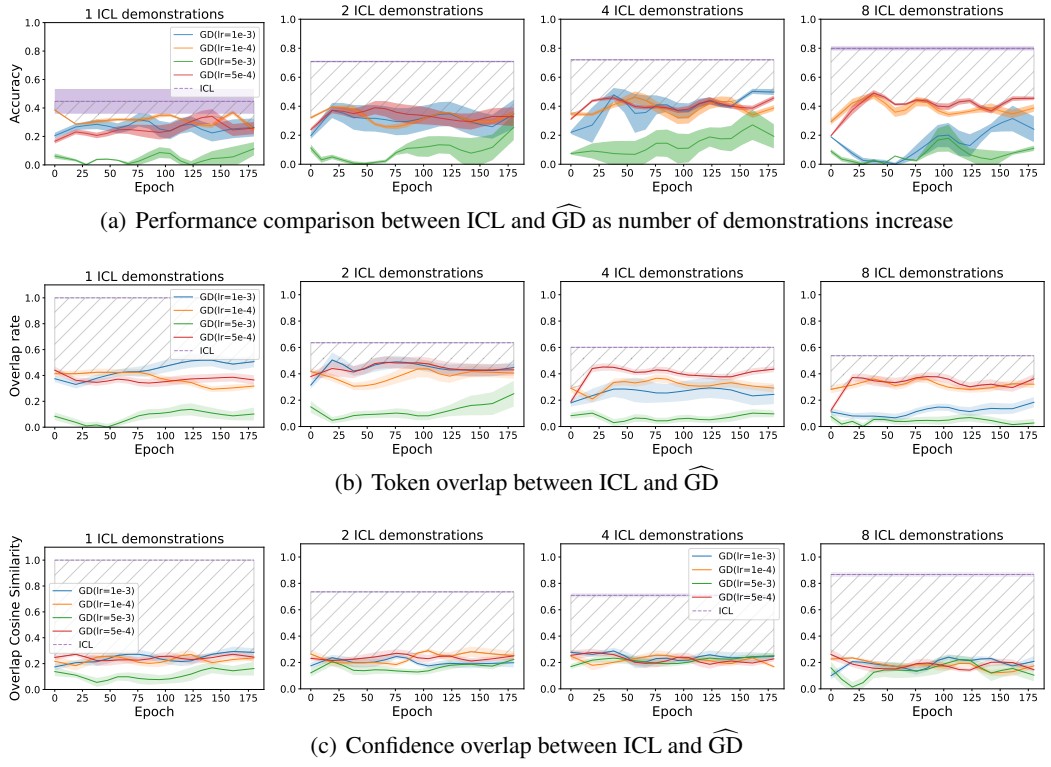

(a) Performance comparison between ICL and $\widehat{\text{GD}}$ as number of demonstrations increase

(b) Token overlap between ICL and $\widehat{\text{GD}}$

(c) Confidence overlap between ICL and $\widehat{\text{GD}}$

Figure 20: Comparison of ICL and $\widehat{\text{GD}}$ on our three evaluation metrics for the CB dataset. $\widehat{\text{GD}}$ is simulated by optimizing on one random middle layer of LLaMa.

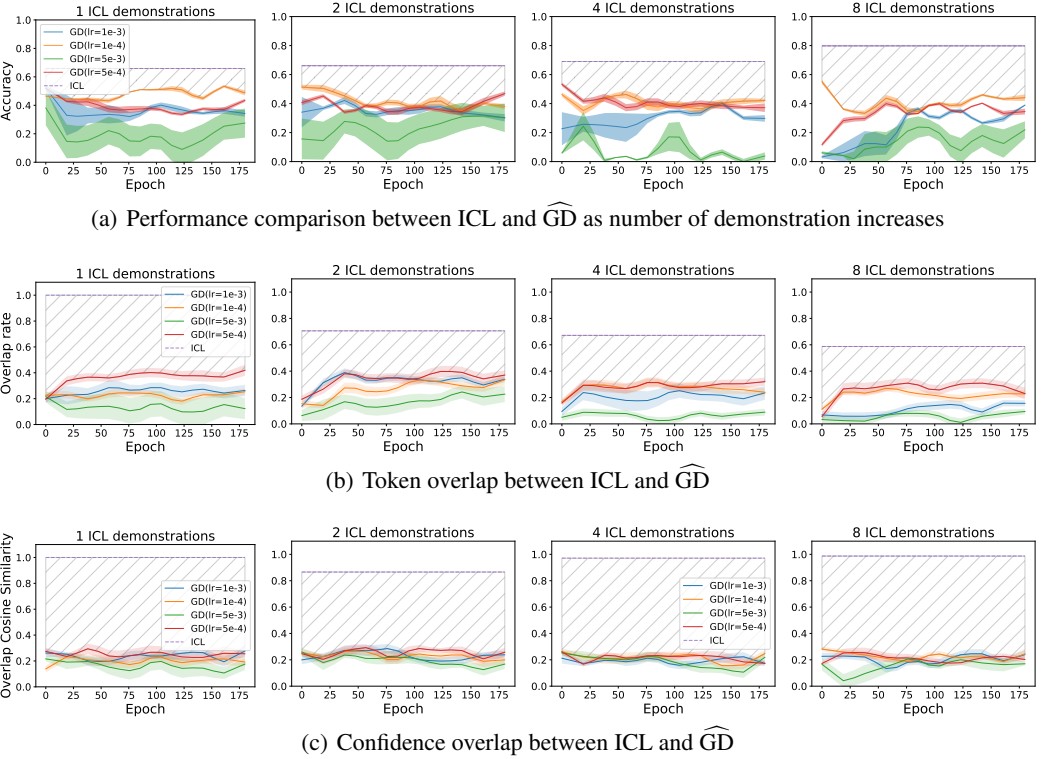

(a) Performance comparison between ICL and $\widehat{\text{GD}}$ as number of demonstration increases

(b) Token overlap between ICL and $\widehat{\text{GD}}$

(c) Confidence overlap between ICL and $\widehat{\text{GD}}$

Figure 21: Comparison of ICL and $\widehat{\text{GD}}$ on our three evaluation metrics for the RTE dataset. $\widehat{\text{GD}}$ is simulated by optimizing on one random middle layer of LLaMa.

**Results of ICL vs. $\widehat{\text{GD}}$ (Multiple layers)** Here are the results for randomly selecting eight layers from LLaMa; we repeat the experiments five times and plot the mean and std. The results are shown in Figure 22 - Figure 25; we can still observe the huge gaps between ICL and $\widehat{\text{GD}}$. Specifically, compared to the 1-layer case, 8-layer simulated $\widehat{\text{GD}}$ achieves higher similarity towards ICL. Moreover, both have lower similarities towards ICL than GD (full-model fine-tuning), which indicates that GD with more parameters brings a higher similarity towards ICL.

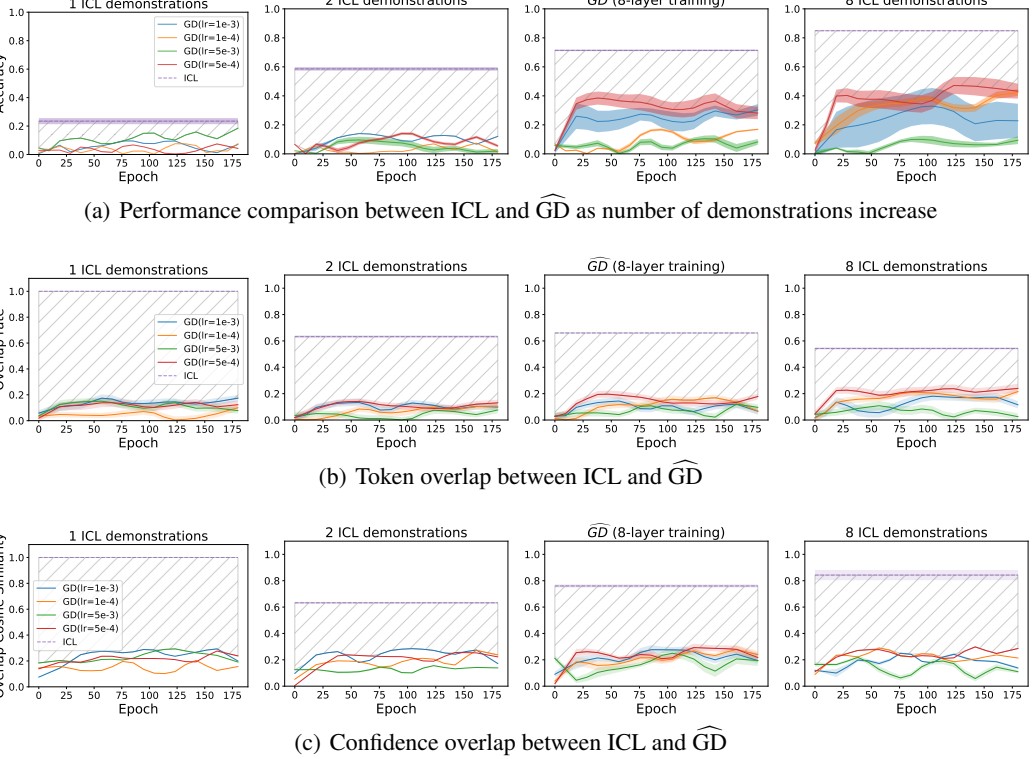

(a) Performance comparison between ICL and $\widehat{\text{GD}}$ as number of demonstrations increase

(b) Token overlap between ICL and $\widehat{\text{GD}}$

(c) Confidence overlap between ICL and $\widehat{\text{GD}}$

Figure 22: Comparison of ICL and $\widehat{\text{GD}}$ on our three evaluation metrics for the AGNews dataset. $\widehat{\text{GD}}$ is simulated by optimizing on 8 layers of LLaMa.

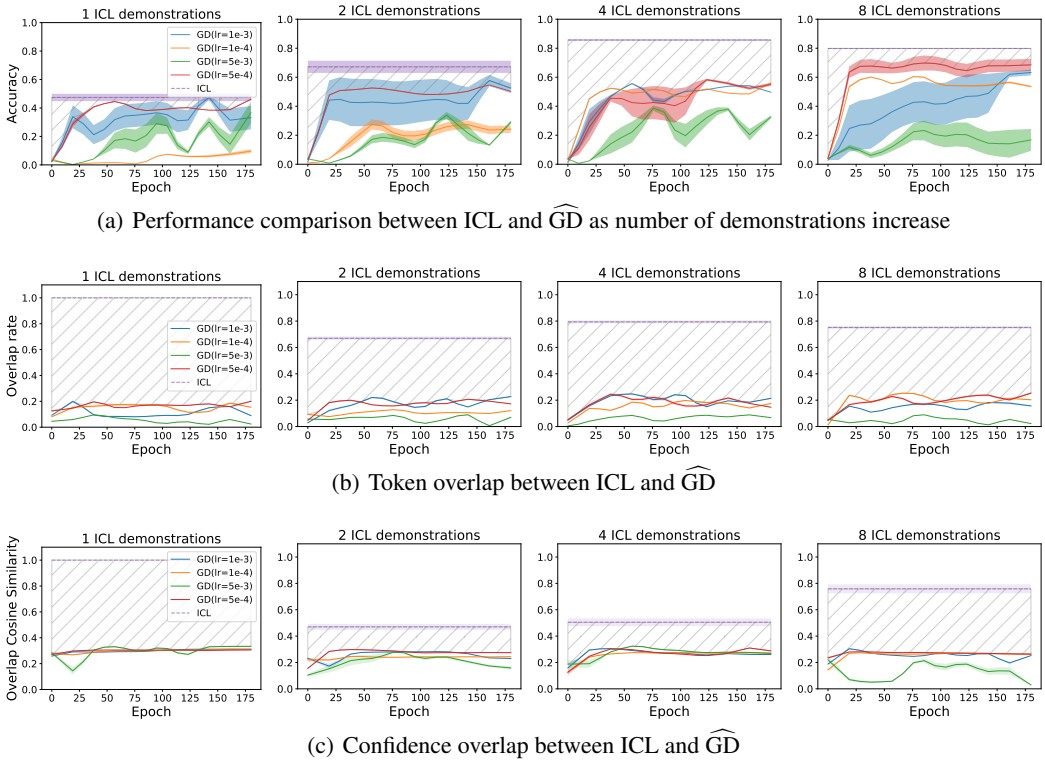

(a) Performance comparison between ICL and $\widehat{\text{GD}}$ as number of demonstrations increase

(b) Token overlap between ICL and $\widehat{\text{GD}}$

(c) Confidence overlap between ICL and $\widehat{\text{GD}}$

Figure 23: Comparison of ICL and $\widehat{\text{GD}}$ on our three evaluation metrics for the SST dataset. $\widehat{\text{GD}}$ is simulated by optimizing on eight random layers of LLaMa.

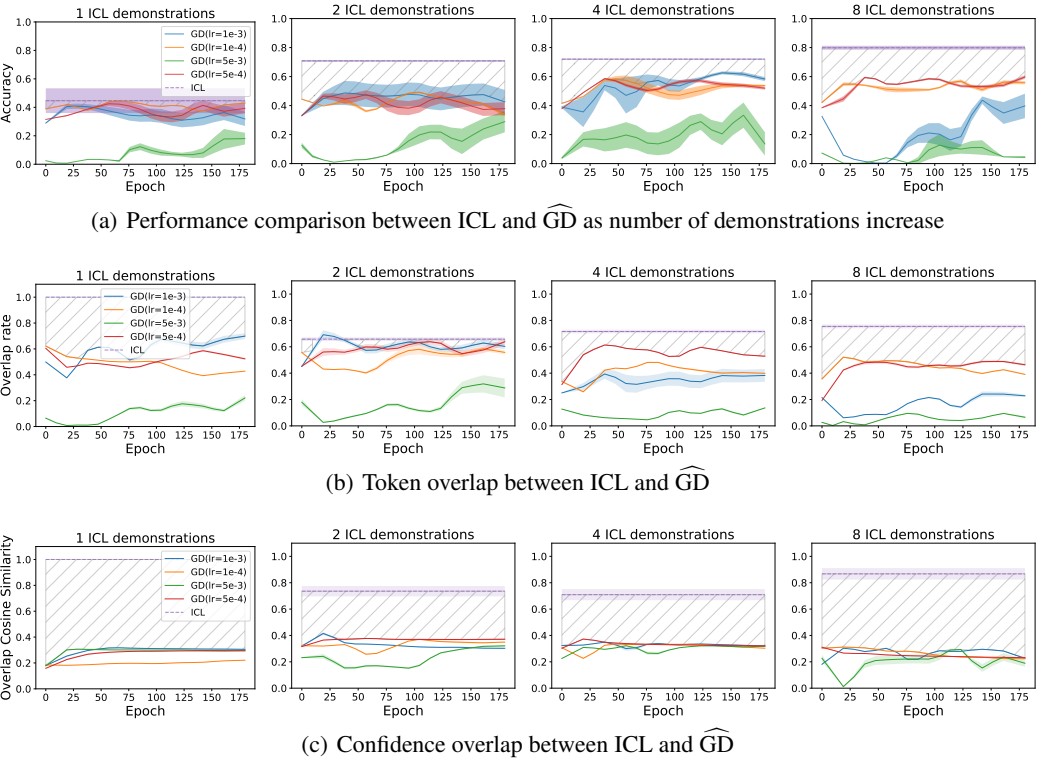

(a) Performance comparison between ICL and $\widehat{\text{GD}}$ as number of demonstrations increase

(b) Token overlap between ICL and $\widehat{\text{GD}}$

(c) Confidence overlap between ICL and $\widehat{\text{GD}}$

Figure 24: Comparison of ICL and $\widehat{\text{GD}}$ on our three evaluation metrics for the CB dataset. $\widehat{\text{GD}}$ is simulated by optimizing on eight random layers of LLaMa.

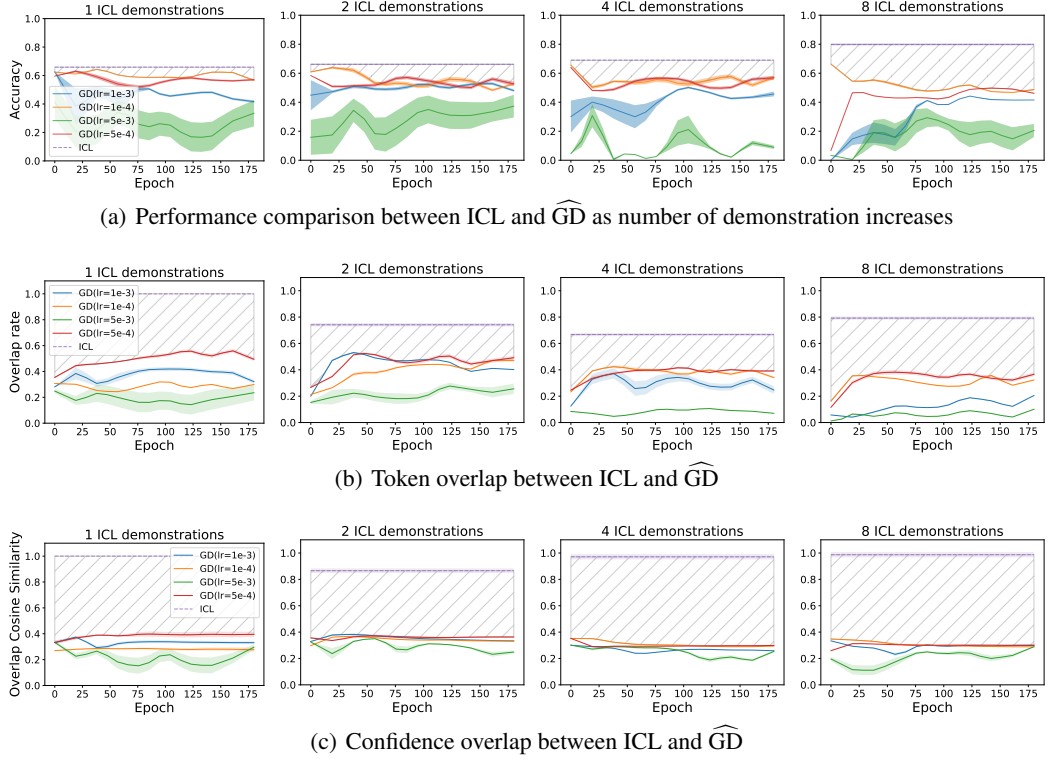

(a) Performance comparison between ICL and $\widehat{\text{GD}}$ as number of demonstration increases

(b) Token overlap between ICL and $\widehat{\text{GD}}$

(c) Confidence overlap between ICL and $\widehat{\text{GD}}$

Figure 25: Comparison of ICL and $\widehat{\text{GD}}$ on our three evaluation metrics for the RTE dataset. $\widehat{\text{GD}}$ is simulated by optimizing on eight random layers of LLaMa.

## G    IMPACT OF MODEL CAPACITY ON THE ICL VS GD.

Next, we investigate the influence of model size on the gap between ICL and GD. Specifically, we fix the number of demonstration size as 8, and select GPT2-XL (Radford et al., 2019), GPT-NEO (Black et al., 2021), GPT-J (Wang & Komatsuzaki, 2021) as models of choice to conduct ICL vs GD experiments. Please note that the model capacity is ranked as follows: LLAMA (7B) >GPT-J (6B)>GPT-NEO (2.7B)>GPT2-XL (1.5B). The results are shown in Figure 26, from where we can see the gap basically does not change significantly as the model changes from GPT2-XL to LLAMA.

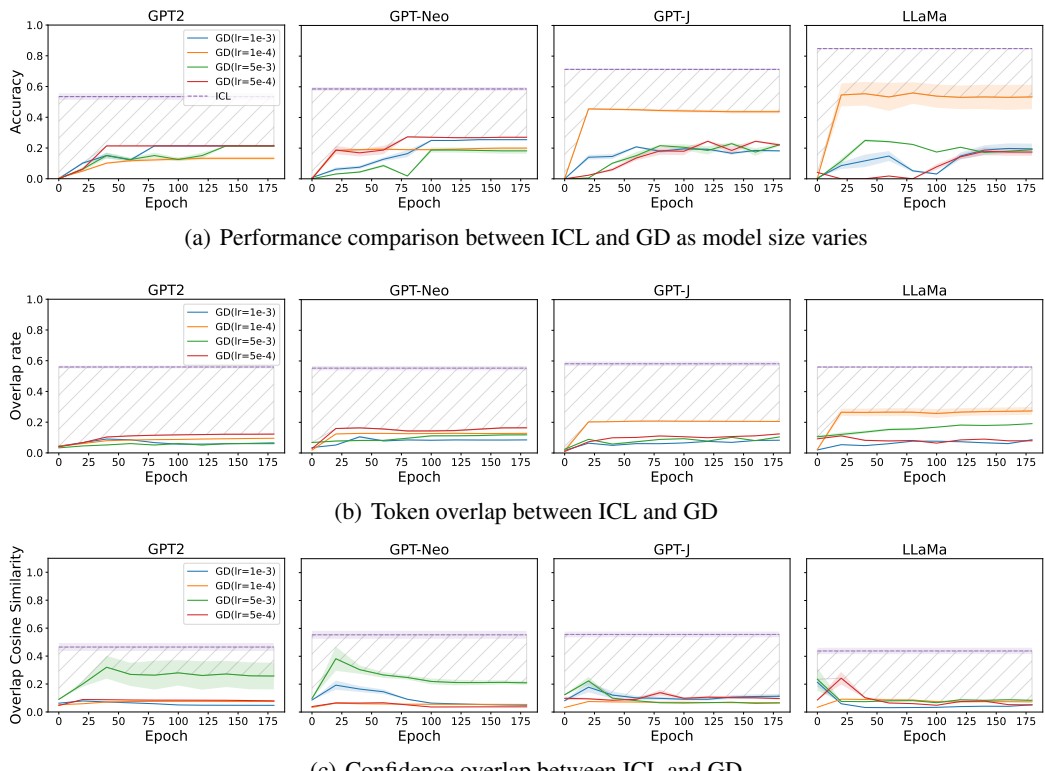

(a) Performance comparison between ICL and GD as model size varies

(b) Token overlap between ICL and GD

(c) Confidence overlap between ICL and GD

Figure 26: Comparison of ICL and GD for the AGNews dataset as model size varies. The substantial gap between ICL and GD (the gray slanted lines) does not change with model size.

## H    RELATED WORK: DISTRIBUTIONAL AND EMPIRICAL EXPLANATIONS OF ICL

**Distributional explanations.**    This body of work explains ICL via distributional frameworks and the relevant properties of LLMs (Xie et al., 2021; Wies et al., 2023). Xie et al. (2021) explain ICL as implicit Bayesian inference, which implicitly maps a given set of demonstrations to an appropriate latent concept (task) learned via pretraining on a massive unsupervised corpus. Similarly, Hahn & Goyal (2023) theorize that natural language pretraining data consists of compositional structure, which leads to the emergent ability of in-context learning, while Chan et al. (2022) show that this might be because of distributional properties of the training distribution (like burstiness). These are all reasonable explanations of how ICL works, although they are somewhat tangential to the focus of this study.

**Empirical explanations.**    Various empirical works study ICL under various settings (Brown et al., 2020; Zhao et al., 2021; Min et al., 2022; Mishra et al., 2022; Han et al., 2023). To note a few, Srivastava et al. (2023) famously benchmarked ICL for many tasks and models. Perez et al. (2021); Lu et al. (2022) showed the sensitivity of ICL to the choice of demonstrations and their orderings. Shin et al. (2022); Razeghi et al. (2022) show the sensitivity of ICL performance to the frequency

and size of the relevant pretraining corpus. Pan et al. (2023) disentangle task recognition and task learning in ICL. These works highlight numerous ways the ability of models to perform ICL changes under different conditions but do not explain how it functions.

