# OpenReview forum: "Do Pre-trained Transformers Really Learn In-context by Gradient Descent?"
_ICLR.cc/2024/Conference — Submitted to ICLR 2024_

### Official Review · Reviewer_pVVn · 2023-10-30

**Soundness:** 2 fair
**Presentation:** 2 fair
**Contribution:** 2 fair
**Rating:** 3
**Confidence:** 3

**Summary:**

The paper presents arguments against the hypothesis that in context learning ICL emulates gradient descent in trained transformers. The arguments can be listed as follows:
* The objective considered in [0] is regression and "is very different from how real language models are trained are trained on the [causal language modeling] objective."
* Gradient descent is agnostic to the sample order in the batch. ICL is not agnostic to the sample order in the batch.
* The construction of the transformer in~\ref{} is contrived.

The paper also presents additional empirical evidences with Llama-7b to support their argument. In summary, they argue that fine-tuning the model does not lead to model outputs that are equivalent to the one obtained with in context learning.

[0] https://arxiv.org/pdf/2212.07677.pdf

**Strengths:**

The paper challenges an emerging theory that transformers learn In Context Learning by gradient descent, which may help bridging the gap between the theory and practical observation. The authors make an effort to be formal about the definitions put forward in the paper, helping in clearing up some of the ideas put forward.

**Weaknesses:**

**The paper presents several arguments with little substance and cohesion.**
For example, the authors claims that the because [0] made their analysis with linear regression then it cannot be comparable to a model trained with causal language modeling. This claim can both be true or false. For example, one could argue that training with linear regression leads to exactly the same solution than a model trained with causal language modeling under certain condition. The authors should prove that the model trained with linear regression leads to a different solution than the model with causal language modeling and not merely state it.

**The authors make several arguments that do not prove the main thesis**
My understanding of the main thesis of this work is that the ICL setup presented in [0] is not equivalent to gradient descent. However, the setup in [0] considers linear self-attention and not general transformers trained with causal masking. To make their point, the authors should explicitly say what is wrong in the work or setup of [0].

If what the authors is trying to say is that the setup of [0] is contrived, then the thesis is not very surprising or significant as no one would be astonished to learn that linear self-attention and hand crafted parameters are unrealistic setups.

Minor:
* The font of some of the figures is a bit too small making it hard to read.
* The font across the figures and the text is not consistent.

Finally, I would like to encourage the authors to revisit the style of their article. While reading their work, I found the writing to be adversarial against an emerging line of work, which could potentially turn out to be, at least partially, true. Instead, having a constructive writing where they, for example, build on top of the existing theory or correct part of the theory would be more enjoying for me to read than a paper that tries to prove a line of work to be wrong.

[0] https://arxiv.org/pdf/2212.07677.pdf

**Questions:**

I gave most of my comments/suggestions in the weakness section.

* Under what conditions does a model as considered in [0] does not lead to transformers that learn in-context by gradient descent?
* Prove that a model as considered in [0] does not lead to transformers that learn in-context by gradient descent.
* What elements of [0] leads to a contradiction that models as considered in [0] does not lead to transformers that learn in-context by gradient descent?

[0] https://arxiv.org/pdf/2212.07677.pdf

---

> ### Author Response · Authors · 2023-11-21
> **Rebuttal**
>
> We thank the reviewer for their review and request them to see **our general response highlighting the changes that were made to the initial draft**.
>
> > The authors make several arguments that do not prove the main thesis.
> > the authors should explicitly say what is wrong in the work or setup of [0].
>
> Please refer to our general response that clarifies our main thesis. As we clarify in the general response, our intention is **not** to prove the existing line of work wrong. Rather, we are studying the curious problem of whether the existing results stretch beyond the boundaries of their assumptions. This is clarified in our updated draft and we request the reviewer to take a look at it.
>
> > ... authors claims that the because [0] made their analysis with linear regression then it cannot be comparable to a model trained with causal language modeling ...
>
> We would like to clarify the difference between ICL and CLM objectives which has been detailed in our **updated section 3.1**. “CLM objective” refers to training of transformers on _task-agnostic_ data. This general-purpose data does not specifically train transformers to perform ICL on a narrow family of tasks. In contrast, “ICL objective” refers to training of transformers exclusively and explicitly on the family of tasks, on which ICL is evaluated later. To gain insights into how ICL is achieved by LMs, choosing a simpler family of tasks like linear regression for analysis is perfectly reasonable, as long as the learning setup remains analogous to real world setups so that the conclusions can be reasonably extrapolated. Therefore, our problem is not with linear regression (the choice of family of tasks), rather how the transformers are trained and the misrepresentation of ICL in that setting.
>
> > …  the authors should prove that the model trained with linear regression leads to a different solution
>
> We have highlighted many fundamental differences between emergent ICL and the studied $\widehat{\text{ICL}}$ in previous works. We believe that these differences are reasonable evidence that they’re likely not the same. If one claims that they are indeed the same, we believe they should prove it.
>
> > the style of their article …  adversarial against an emerging line of work
>
> Thanks for sharing this concern. We certainly did not mean to appear adversarial to the prior work or dismiss them – these are excellent and innovative works. At the same time, the community in our academic discourse would benefit from added clarity on the implications of these works, which we worry may be misunderstood.
>
> In our revision, we have tried to revise all the statements so that they’re objective and not over-claiming. If you feel that there is a place that would benefit from further precise statements, **please let us know**. Thank you!
>
> > font of some of the figures, consistency
>
> We have tried to make the font larger and more consistent. Please let us know if anything else catches your eye.

---

> > ### Author Response · Authors · 2023-11-22
> >
> > Dear Reviewer — We have considered your valuable suggestions and have made revisions in our manuscript (marked in purple). If you have additional questions or feedback, do not hesitate to reach out to us. If our response has effectively addressed your issue, we would appreciate it if you could consider raising the rating. Thank you!

---

> > > ### Author Response · Authors · 2023-11-23
> > >
> > > Dear Reviewer,
> > >
> > > We have put so much effort into addressing all comments from all reviewers to revise the draft. We would deeply appreciate if you could comment on what still feels not convincing. We hope that you can raise the score if no such issues remain.

---

### Official Review · Reviewer_ivcd · 2023-10-31

**Soundness:** 3 good
**Presentation:** 4 excellent
**Contribution:** 2 fair
**Rating:** 6
**Confidence:** 4

**Summary:**

The paper conducts a deeper study into the hypothesis of in-context learning in LLMs as a simulation of gradient descent on an auxiliary model. The authors formalize a few sets of functional properties and compare in-context learning (ICL) and gradient descent (GD) on these properties.  The authors observe inconsistencies between the two algorithms in different settings and hence provide empirical evidence against the equivalence of ICL and GD in the realistic setting.

**Strengths:**

The paper is well-written and the logic is easy to understand. The authors mention the functional properties with which they compare in-context learning and gradient descent on a large LLM model. In their experimental study, instead of simply comparing the two algorithms in terms of performance, the authors use metrics like token overlap and cosine similarity to show a clear distinction between the two algorithms. Overall, this paper provides an extensive study pointing out the differences between the two algorithms and provides clear insights into how the community can redesign the existing hypothesis for realistic model settings.

**Weaknesses:**

I have a few questions on the experimental study. Please find them below.

(a) **Hypothesis under study in section 4:**  In this section, the authors assume equivalence between ICL and GD on the same model. However, Akyurek et al.'22, and Oswald et al'23 argue that transformers train a small auxiliary model (different from the parent transformer) inside. Hence, the setting that the authors consider aligns more with the result of [1], which claims in-context learning as implicit optimization of the same parent model. So, if I understand currently, the authors are simply refuting [1]'s hypothesis. Can the authors comment on this discrepancy? If the authors want to refute the hypothesis completely, then maybe they need to search for all possible sets of auxiliary models that can fit inside.

(b) **Arguments against sparsity:** I believe, the argument that trained transformers are not sparse, as given by the constructions of Akyurek et al.'22, and Oswald et al'23, isn't a valid argument for the discrepancy between the two algorithms because the previous works simply aim to give an expressivity result on transformers. It is certainly possible that transformers find a dense and more compressed solution to simulate gradient descent. To completely refute the argument, the authors need to refute the probing experiments that the previous works did to search for traces of gradient descent inside these trained models (which I believe is a herculean task).

Furthermore, instead of simply looking at the movement of the transformer weights across training to argue that the model doesn't stabilize to a single sparse solution, maybe the authors can come up with experiments to suggest that the model changes its internal mechanism across training, instead of simply using different weight matrices to represent the same internal mechanism across time (which again is a herculean task).


(c) In section 4, the model has been fine-tuned with a cross-entropy loss for GD, with the candidate set being the entire vocabulary. Are the contextual examples concatenated with the test query, like [2]? This setting is more likely since ICL uses demonstrations concatenated, which provides a prior to the right candidate set.

Furthermore, instead of simply optimizing the cross-entropy loss with the entire vocabulary being the candidate set, maybe the authors can put more weight on the relevant logits during training and inference.

(d) What do an overlap rate and cosine similarity of 1.0 mean and < 1.0 mean for ICL in 1 demonstration and >2 demonstration settings in Figure 4?

Overall, I believe the paper attempts to take a deep dive into a very difficult question using simple experiments, which is impressive in itself. However, as far as I understand, these experiments don't refute the hypothesis of equivalence between GD and ICL completely. Instead, they simply ask the community to make small changes to the hypothesis (e.g. SGD in place of GD, auxiliary internal model isn't the same parent model, etc.). Hence, I have a slightly lower score but am happy to discuss it during the rebuttal period.


1: Why Can GPT Learn In-Context? Language Models Implicitly Perform Gradient Descent as Meta-Optimizers. Dai et al'23.

2: Making Pre-trained Language Models Better Few-shot Learners. Gao et al'21.

**Questions:**

Please see my questions in the previous section.

---

> ### Author Response · Authors · 2023-11-21
> **Rebuttal**
>
> We thank the reviewer for their detailed review and request them to see our general response highlighting the changes that were made to the initial draft.
>
> > Hypothesis under study in section 4
>
> > the authors assume equivalence between ICL and GD on the same model.
>
> We have revised our draft so that the connection of Hypothesis 2 with prior work is clearer and we have included experiments on sub-model GD. Section 4 contains experimental results that involve (implicit) sub-models that are more comparable with the setup of Akyurek et al., and Oswald et al., i.e. hypothesis 2. Because we can not search over all possible subsets of the parameters, our experimental setup specifically focused on “intuitive subsets”` that were inspired by the construction of the earlier works: Von Oswald et al. hypothesized implicit model lies in $W_V$ of Transformers, while the probing experiments of Akyürek et al. suggest that this optimization happens in top layers (suggesting that initial layers focus more on representation learning).
>
> We conducted an experiment where we only fine-tuned $W_V$ of a single layer, for different choices of higher layers (with ablations) in the LLaMa model in hopes of fine-tuning only the implicit model. Based on our notation (Figure 1), call this model $\widehat{\text{GD}}$. We then compare $\widehat{\text{GD}}$ to ICL on our three metrics. We found that the _differences between the two remain significant_. We have added this result in the updated draft (Figure 2).
>
> > Arguments against sparsity ... previous works simply aim to give an expressivity result on transformers
>
> As we clarify in the general response, we are curious about the generalizability of these expressivity results beyond the scope of their assumptions. The previous “expressivity” works’ definition of ICL ($\widehat{\text{ICL}}$: our notation) is different from the emergent ICL in pre-trained transformers. Moreover, these works contain experiments and statements that imply **stronger** conclusions.
> - In Von Oswald et al., there are a variety of experiments comparing ICL vs. GD and emphasizing the transformer weights with the hand-constructed weights. They conclude that: `"... when training multi-layer self-attention-only Transformers on simple regression tasks, we provide **strong evidence that the construction is actually found**"`.
> - You can see their stretching of the claims in the title of the published work: “Transformers learn in-context by gradient descent,” though a more accurate statement is “Transformers **have the capacity to** learn in-context by gradient descent”
> - A similar weight comparison (implicit linear weight difference; section 4.1) is done by Akyürek et al, that implies these weights do emerge by training transformers. Although we agree that real models might find some dense counterparts of the simple constructions, more efforts should be made to find those families of weights.
>
> These stronger conclusions, stretching beyond the boundaries of “expressivity,” have motivated our work. We provide a variety of empirical and theoretical arguments that interpreting ICL as [implicit] GD is not a foregone conclusion, and the community should view these results with a grain of salt.
>
> Just to be clear, our intention is not to dismiss prior work such as Akyürek et al., Von Oswald et al., Dai et al., etc. They are important results, but we feel that the community would benefit from additional clarity about their implications.

---

> ### Author Response · Authors · 2023-11-21
> **Continued**
>
> > ... the authors need to refute the probing experiments ...
>
> Thanks for raising this question. First, we are **not** claiming to refute the theoretical formalism of Akyürek et al. However, taking their conclusions beyond the scope of their expressivity results needs extreme care.
>
> As you hinted, Akyürek et al. use their “probing” experiments (Section 5 of their work) to show that a transformer's internal representations can be used to predict parameters or quantities that are necessary for implicit (internal) iterative optimization.
>
> This cannot be taken as proof, for several reasons:
>
> - The existing “probing” techniques are known to be problematic as they **conflate “causation” vs. “correlation”**. As the literature suggests, `”...the probing framework may indicate correlations between representations fl(x) and linguistic property z, but **it does not tell us whether this property is involved** in predictions of f”` (Belinkov 2022).
> - Even if we take their result as causal evidence, the probing experiment is done on quantities ($X^TY$), which could be used to perform **many** other algorithms (GD or otherwise). The question should not be whether a transformer representation can predict a solution, but how it is computed (the functional nature of the algorithm) – which is not covered by the probing setup.
> - Last but not least, the authors use a Transformer that is **already trained with ICL objective**: `“We take a trained in-context learner, freeze its weights, then train an auxiliary probing model”`. Using the ICL objective has significant inductive biases (about the structure of the sequence it sees) which is different from real LLM training.
>
> Belinkov, Yonatan. "Probing classifiers: Promises, shortcomings, and advances." Computational Linguistics 48.1 (2022): 207-219.
>
> > ... come up with experiments to suggest that the model changes its internal mechanism across training ...
>
> We do **not** claim that ICL is equivalent to any mechanism (GD or otherwise). With our evolution of weights result, we just want to highlight that the ability of LMs to perform ICL remains stable even with evolving parameters. This just means that any claim of equivalence should be with a mechanism that elicits a family of weights, rather than fixed sparse constructions.
>
> > Are the contextual examples concatenated with the test query…
>
> We concatenate each ICL demo, including the test query, with ‘\n,’ which is similar to Gao et al'21. (they apply [MASK] in ICL for BERT).
> Our choice to fine-tune the model with a cross-entropy loss over the entire vocabulary was deliberate. This decision aligns with standard fine-tuning practices. While it's valid that placing more weight on relevant logits during training and inference could potentially refine the model's predictions, our primary goal was to evaluate the models in a setting that closely mirrors how they are commonly fine-tuned and deployed in real-world applications (for example, most of the works in the “instruction tuning” literature).
>
>
>
> > About overlap rate and cosine similarity?
>
> In the figure, the purple dashed line represents the gap between two instances of ICL that _vary solely in the order of demonstrations_.
>
> When the demonstration number equals 1, the overlap rate and cosine similarity metrics equal 1. This is because there is inherently only one possible order for a single demonstration. On the other hand, multiple orders are possible when the number of demonstrations exceeds one, introducing variability in the model’s output. In Figure 5, we include this comparison to show the impact that varying orders of demonstrations have on the model's ICL performance, as quantified by our three metrics.

---

> > ### Author Response · Authors · 2023-11-22
> >
> > Dear Reviewer — We have considered your valuable suggestions and have made revisions in our manuscript (marked in purple). If you have additional questions or feedback, do not hesitate to reach out to us. If our response has effectively addressed your issue, we would appreciate it if you could consider raising the rating. Thank you!

---

> > > ### Comment · Reviewer_ivcd · 2023-11-22
> > >
> > > I am sorry for the delay in my reply. I thank the authors for their detailed rebuttal. I certainly agree with the authors about the results being stretched beyond expressivity in multiple previous works regarding ICL vs GD equivalence. Hence, I liked the idea of disproving equivalence between the two algorithms that this paper aims to achieve. Furthermore, I have always had the same concerns about the differences between $\hat{\text{ICL}}$ and ICL that the authors cleanly report in this paper.
> > >
> > > I still have concerns about the exact hypothesis that the paper aims to verify in the experiments. It is difficult to find the exact implicit model inside the transformer and tune it with a learning algorithm that a transformer might use to tune on the fly. Previous papers show the expressivity of transformers to be able to internally tune small models using gradient descent implicitly. It's difficult to pinpoint how the model does this in practice. According to me, the current paper aims to remove the simplistic constructions that we can think of.
> > >
> > > I am happy to discuss more on this.

---

> ### Author Response · Authors · 2023-11-22
>
> Thanks for going through our detailed rebuttal!
>
> > ... concerns about the exact hypothesis that the paper aims to verify in the experiments
>
> As the reviewer already understands, previous works' claims are about the equivalence of $\widehat{\text{ICL}}$ and  $\text{GD}$/$\widehat{\text{GD}}$. What we wanted to test with our experiments was whether this implies any equivalence between $\text{ICL}$ and $\text{GD}$/$\widehat{\text{GD}}$. Therefore, we performed two sets of experiments (Section 4) for comparing $\text{ICL}$ with $\text{GD}$ and $\widehat{\text{GD}}$.
>
> 1. In practice, $\text{GD}$ is implemented by fine-tuning the whole model on task-specific data, which is what we did. We found that _$\text{ICL}$ and $\text{GD}$ behaved differently on our three evaluation metrics aimed at looking at functional equivalence_. In fact, Dai et. al.  in their experiments look at the same setting (whole model tuning) but only base their equivalence on raw performance which does not paint the whole picture about functional equivalence.
> 2. For $\widehat{\text{GD}}$, because it is hard to find where exactly would this hypothetical implicit model lie in a big "transformer" like LLaMa, we relied on the suggestion of reviewer **uBmS** to look for "intuitive subsets" of the model to consider as implicit models. We _based our intuition on previous works which hypothesize the existence of these implicit models_. Particularly, we used von Oswald et al.'s construction, which says that the implicit model lies in $W_V$ of the transformer, and Akyürek et al.'s probing experiments to guide us about which layers we should be looking at (deeper layers). Using these intuitions, we performed $\widehat{\text{GD}}$ updates in three different studies:
>    - Fine-tuning $W_V$ in one of the deep layers (randomly choosing one of the last 4 LLaMa layers).
>    - Fine-tuning $W_V$ in one of the middle layers (randomly choosing one of the layers in layers 16-20 of the 32 in LLaMa).
>    - Fine-tuning all weights in 8 layers randomly selected out of the 32 in LLaMa.
>
> In our updated section 4, we have presented results from all these experiments for $\text{GD}$ and our intuitive understanding of $\widehat{\text{GD}}$ when compared with $\text{ICL}$ and found that they _function differently in all cases_.
>
> We agree with the reviewer that these comparisons for $\widehat{\text{GD}}$ are not exhaustive and do not invalidate the claim of equivalence completely, but as mentioned in our future work section, identifying implicit models used for GD in LLMs in a computationally feasible manner could be an interesting avenue of future research.

---

> > ### Comment · Reviewer_ivcd · 2023-11-22
> >
> > Thanks for the detailed response.
> >
> > I agree with the authors that their paper poses an important question to the community: "Identifying implicit models used for GD in LLMs in a computationally feasible manner."
> >
> > I like the message of the paper. I still have concerns about the rigor of the experiments and the hypothesis that the paper aims to disprove. I am slightly increasing my score. I will discuss it further with my fellow reviewers during the discussion period.

---

> > > ### Author Response · Authors · 2023-11-23
> > > **Rebuttal**
> > >
> > > We thank the reviewer for reading our responses and raising their score. Your feedback is super useful for making our work clearer.
> > >
> > >  > “the experiments and the hypothesis that the paper ”
> > >
> > > While we laid out the thesis of the work in the general response, here we try to present it differently. For exposition, we’re quoting the two hypotheses discussed in the introduction:
> > >
> > > -  **H1:** "For any Transformer weights resulting from self-supervised pretraining and for any well-defined task, ${\text{ICL}}$ is algorithmically equivalent to GD (whole model or sub-model)."
> > >
> > >  - **H2:** "For a given well-defined task, there exist Transformer weights such that $\widehat{\text{ICL}}$ is algorithmically equivalent to GD (whole model or sub-model)."
> > >
> > > Most existing work made claims that fit in **H2**. But the way the results are reported, they can be misinterpreted as **H1**. This naturally begs the question: Does **H2** $\implies$ **H1**? That is the central hypothesis of this study.
> > >
> > > Our figure 1 shows this connection to hypotheses (H1 corresponds to circled (A)/(B) while H2 corresponds to (C)). **Note that we have updated Fig1 a bit. Please use our latest version.**
> > >
> > >  - To assess (A) in LLMs, some works (like Dai et. al.) compare the raw performance of ICL vs fine-tuned models. We argue with our fine-grained evaluation metrics that these comparisons do not paint the whole picture and the functional behavior of the two approaches are still different (results in section 5). We also make theoretical arguments against this by our order-sensitivity argument in section 4.
> > >  - To assess (B) in LLMs, we have tried “intuitive choices” of sub-models (inspired by previous works’ constructions) and showed their results in section 5, but it still remains an open question. However, our arguments about order-sensitivity in section 4, still apply in this setting.
> > > - We discuss (C) in section 3. Recent works aim to answer this question by showing hand-constructed weights that can perform GD on a given sub-model. However, this treatment of in-context learning is far from the realistic settings (i.e., emergent ICL in LLMs). We show the limitations of this setting in section 3.1, and argue about the scalability of hand-constructed weights in section 3.2. Moreover, our results from section 4 cast doubts on the stretched implications of these works about **H1**.
> > >
> > > We hope that this explanation clarifies the thesis of our paper. Again, we appreciate your feedback on what is unclear or what should be improved.

---

### Official Review · Reviewer_GeTu · 2023-10-31

**Soundness:** 2 fair
**Presentation:** 2 fair
**Contribution:** 2 fair
**Rating:** 5
**Confidence:** 2

**Summary:**

Recently, connections have been built between in-context learning (ICL) and gradient descent (GD), in order to better understand in-context learning. This paper challenges such connections from both theoretical and empirical perspectives. Specifically, the authors establish the difference between ICL and GD in terms of order sensitivity and demonstrate that the assumption regarding model parameters for the connection hardly holds in practice. Further, the paper proposes metrics to empirically evaluate to what extent ICL and GD perform differently.

**Strengths:**

1. The paper focuses on the topic of how to understand in-context learning, which is crucial to the field.

2. The proposed perspective of order sensitivity to look into the difference between ICL and GD is interesting and looks novel to me.

**Weaknesses:**

1. I am a bit worried about the significance. The proposed order sensitivity is interesting and yet requires more in-depth analysis (see questions). However, I am not an expert in this field so I will defer to other reviewers regarding this point.

2. The writing can be improved.  E.g., there are typos such as "We know that both if ..." and "This is a relative metric is computed based...".

**Questions:**

1. As the author mentioned, the construction of Akyurek et al. ¨ (2022) allows for order sensitivity in GD by update on samples one by one. Do we still have the difference in terms of order sensitivity in that setting?

2. What if we use specific types of positional encoding to make ICL agnostic to the order of demonstrations? Would the performance increase or not?

3. Alternatively, we can use the average prediction of many random orders of demonstrations to make ICL agnostic. Is that setting explored?

---

> ### Author Response · Authors · 2023-11-21
> **Rebuttal**
>
> We thank the reviewer for their time to review our paper and request them to **look at our general response, which highlights changes** made to the initial draft of the paper to improve its structure.
>
> > The proposed order sensitivity is interesting and yet requires more in-depth analysis ...
>
> Beyond the comparison of the order sensitivity of ICL vs GD, in the appendix, we had results on the order sensitivity of variants of GD (like SGD and Adam) which provide a broader context. As suggested by another reviewer, we have brought these to the main text.
>
> We also did ablations on the SGD comparison with varying batch sizes (added in the appendix of the updated draft), although it is unclear how the model would decide which order to use in-context examples in the formation of Akyürek et al., our comparisons show that _ICL is still substantially more sensitive to order than SGD or Adam_.
>
> > What if we use specific types of positional encoding to make ICL agnostic to the order of demonstrations?
>
> We want to clarify that our objective with this paper is **not** to make ICL order-stable (although that’s an interesting problem to focus on). ICL does **not** necessitate order-stability as it emerges in real-world models like LLaMa or GPT3 that are shown to be sensitive to input order.
>
> The LLaMA model (which we use for our experiments) uses relative (specifically, rotary embedding (RoPE, Su et al. 2022) positional embedding, which tends to be more robust to reordering context content. However, as our results in section 4 show, they remain order-sensitive. Whether there will be any future way of encoding information that is robust to order is an interesting goal for future research.
>
> We review the key thesis of our work in the general response. Briefly, our work is motivated by curiosity about the generalization of the prior results (on an implicit equivalence between ICL and GD) to more realistic scenarios. To present our case, we show various forms of evidence that may indicate that ICL and GD (on full model or sub-model) work differently. For instance, the high variance of ICL performance wrt order permutations _rather highlights it as a functional difference between ICL and GD_.
>
> > ...  we can use the average prediction of many random orders ...
>
> We note that our results, shown in section 4, have all been averaged across multiple runs where demonstrations and their order are randomly sampled. This is to make sure that any order-sensitivity behavior is not a one-off incident but average behavior over multiple runs.
>
>
> > The writing …
>
> We regret that the writing of the paper was not up to the mark in the initial draft. We have fixed the mistakes the reviewer pointed out and hope they find the revised version much improved.

---

> > ### Comment · Reviewer_GeTu · 2023-11-21
> >
> > Thank you for your response. After reading all the review comments, I tend to keep my rating unchanged.

---

> > > ### Author Response · Authors · 2023-11-21
> > >
> > > We have gone to a great length to update the draft and address your comments. It would be tremendous feedback to us if you can share why you're not happy with the changes. Thank you!

---

> > > > ### Author Response · Authors · 2023-11-23
> > > >
> > > > Dear Reviewer,
> > > >
> > > > We have put so much effort into addressing all comments from all reviewers to revise the draft. We would deeply appreciate if you could comment on what still feels unconvincing. We hope that you can raise the score if you do not confidently feel that any such issues remain.

---

### Official Review · Reviewer_uBmS · 2023-11-01

**Soundness:** 3 good
**Presentation:** 3 good
**Contribution:** 3 good
**Rating:** 6
**Confidence:** 4

**Summary:**

This paper scrutinizes the strong claims that LMs implement gradient descent in inference time to achieve their ICL functionality, and assesses whether specific constructions of such LMs are feasible. The claim that LMs implement internal GD to do ICL is discarded by showing that ICL in LMs are order-sensitive, and ICL output distribution is different from a GD trained model’s distribution. The claim that LMs implement internal GD in a specific construction (e.g. as in Von Oswald et al. or Akyürek et al.) is discarded by showing that the sparsity of LLama LM’s weights significantly less than sparsity of the proposed constructions.

**Strengths:**

1) The authors conducted an extensive set of experiments to compare ICL to fine-tuning a model with GD by comparing order-sensitivity, learning curve of two algorithms. They also compare the token overlap of the resulting predictors.
2) Their results show a clear difference between ICL of LLama model vs fine-tuning LLama model with the same few-shot statsets.
3) The authors also investigated parameter structure of LLama model and showed that it is far from constructed models in (Akyürek et al., and Von Oswald et al.,)

**Weaknesses:**

I think in general the paper needs to go over argumentations. I read both Akyürek et al. and Von Oswald et al. very carefully and here are my issues with the current version of the paper.

1) The GD is an ambiguous algorithm and it is unlucky to be GD in the title of Oswal and hence propagated to this paper. And this has important implications for the experiments presented in this paper. A proper learning algorithm for GD can be specified together with a loss function and a neural network. So, I believe the proper claim to refute should be in the form of “LMs implement  internal GD on X neural network with Y loss function to achieve ICL for all X, Y”  (some X, Y can also be meaningful but doesn’t refute the possibility fully)

2) **Definition-1 and its relation to the strong claim:** The strong claim **cannot be** “LMs implement GD on cross-entropy on themselves” because the Transformer can only implement internal algorithms on a strictly smaller model. For example, Akyürek and Von Oswald show Transformers can implement GD on a linear model way smaller than the actual Transformer that does ICL. However, In Figure-1, authors compare ICL on the same model to GD on the same model with cross-entropy which is inherently impossible to be equal. The same issue of evaluating ICL of a model to GD of the same model exists in Token Overlap experiments as well.  And all of these issues arise from Definition-1 which seeks for equivalence of ICL with some fine-tuned version of the same model.

On the other hand, authors also proposes Definition-2 which is the proper version of the strong claim, however, do not present experiments where only some parts of an LM is finetuned. This unfortunately requires a search over what parameters to finetune which might be computationally expensive. But authors can search over intuitive subsets of all possible parameters.

3) The GD part of the claim is a bit strong to be meaningful. For example, order sensitivity experiments are not related to **SGD** (online GD with some batch size < number of examples) which you also mentioned  in the end of P4 *“... construction of Akyürek allows for in GD as the update is performed on samples one-by-one instead …”*. A better experiments is to look at order-sensitivity of SGD. Those experiments left to Appendix, I suggest moving them to body and displaying together with GD.

4) Akyürek et al. does not make the strong claim that LMs implement GD to achieve ICL, and does not even imply. The paper argues that Transformers can discover learning algorithms to achieve ICL if it’s trained for ICL.

Their main result is that the size of the Transformers changes the learned internal algorithm to achieve ICL. Smaller models are more close to SGD whereas large models implement Bayes optimal **Ridge Regression** solution to the linear regression problem. Even if I assume this paper implies something, it cannot be GD from reading these results. Because it suggests Bayesian learning, we expect ICL to be have a prior or a regularization that is learned during training time.

On the other hand, yes, Von Oswald et al. implies the strong claim in their intro “We find and describe one possible realization of this concept and hypothesize that the in-context learning capabilities of language models emerge through mechanisms similar to the ones we discuss here.”

**Questions:**

- Does the sparsity ratio changes from layer to layer of GPT-J?
- Why do you switch between models GPT-J vs LLama?
- In SGD experiments in the appendix:
  - Did you try different mini batch sizes?
  - Did you shuffle the examples or iterate in the same order?
  - Did you do one pass SGD or multiple?

**Summary of the Review**

Overall, I find the GD vs ICL experiments interesting and highlighting that the community still needs better explanations for ICL. However, I find that the experiments do not refute some of the claims that the authors want to refute (W2), and (W1, W3, W4) important to be addressed before publication. I am hoping to raise my score if these weaknesses can be fixed or answered.

---

> ### Author Response · Authors · 2023-11-21
> **Rebuttal**
>
> We thank the reviewer for their comprehensive insights and request them to **look at our general response,** which highlights what changes were made to the initial draft of the paper.
> 1. **The GD is an ambiguous algorithm ...**
> > …the proper claim to refute should be …
>
> We are **not** suggesting that hypothesis 1 is what should be or can be proven. There are two hypotheses in our paper, one which is a more general notion of equivalence between ICL and GD, while the other is what previous works look at. We are just investigating whether the second implies the first in any way. In our general response and the updated draft, we have made the distinction very clear and request the reviewer to kindly read it.
>
> 2. **Definition-1 and its relation to the strong claim**, about experiment with sub-models (intuitive subsets of the parameters)
>
>   > …authors compare ICL on the same model to GD on the same model with cross-entropy, which is inherently impossible to be equal…
>
>   > ... do not present experiments where only some parts of an LM is finetuned …
>
> As the reviewer pointed out, it is infeasible to search over all subsets of sub-models to find if one corresponds to the hypothesized implicit model. At their suggestion, we have added experiments involving `“intuitive subsets” in Section 4. (Thank you for the suggestion!)
>
> Our experimental setup specifically focused on subsets that were inspired by the construction of the earlier works: Von Oswald et al. hypothesized implicit model lies in $W_V$ of Transformers, while the probing experiments of Akyürek et al. suggest that this optimization happens in top layers (suggesting that initial layers focus more on representation learning).
>
> We conducted an experiment where we only fine-tuned $W_V$ of a single layer, for different choices of higher layers (with ablations) in the LLaMa model in hopes of fine-tuning only the implicit model. Based on our notation (Figure 1), call this model $\widehat{\text{GD}}$. We then compare $\widehat{\text{GD}}$ to ICL on our three metrics. We found that the _differences between the two remain significant_. We have added this result in the updated draft (Section 4).
>
> 3. **The GD part of the claim is a bit strong to be meaningful**
> > ... order sensitivity experiments are not related to SGD ...
>
> We looked at the order-sensitivity for vanilla GD because that is what is studied in Von Oswald et al. We agree with the reviewer’s suggestion about highlighting the sensitivity study with SGD and Adam to show that even if Akyürek et al. allow for order sensitivity, ICL exhibits a lot more sensitivity than SGD/Adam. We have updated the order sensitivity figure (**Figure 2**) in the main text.
>
>
> 4. **About the claim of Akyürek et al.**
> > Akyürek et al. does not make the strong claim ...
>
> While Akyürek et al. are more precise in their claims, there are phrasings that may lead to confusing conclusions of hypothesis 1. Most importantly, their use of ICL (training Transformers with ICL objective) is critically different from the emergent ICL in Transformers (the CLM objective on natural text). These two objectives elicit different families of transformer models ($\text{ICL}$ vs $\widehat{\text{ICL}}$, in our notation in Fig.1), as pointed out in our *updated Background (section 2)*.
>
> The misleading nature of conclusions is in fact evident in your own phrasing. As you suggested, Akyürek et al. argue `“that Transformers can discover learning algorithms to achieve ICL **if it’s trained for ICL**”`. The key assumption *“... if it is trained for ICL”* is easy to miss. Furthermore, if a model “is trained for ICL” is it really capable of ICL?! There is no work that shows the equivalence of this to the emergent ICL in real-world models like LLaMa. We have tried to clarify this distinction in our revised description in “background”. This is one of the first works that adopted the use of ICL objective and its framework has since been used in the several follow-up works that blindly equate $\widehat{\text{ICL}}$ with $\text{ICL}$.
>
> Moreover, even Akyürek et al. seem to over-extend their conclusions' scope. For example, their section 4, which `“aim[s] to explain ICL at the computational level by identifying the kind of algorithms to regression problems that transformer-based ICL implement,”` is premised by saying that `“it is these iterative algorithms that capture the behavior of **real learners**”`, for which we did not find any concrete evidence. Or the title their published work is: “What learning algorithm is in-context learning?” though a more accurate title is “What learning algorithm **can be simulated** in-context by Transformers?”
>
> Just to be clear, our intention is **not to dismiss** prior works. These are excellent and innovative works. Our hope is to bring in more clarity between the actual claims and the extend to which they can be generalized beyond their assumptions.

---

> > ### Comment · Reviewer_uBmS · 2023-11-21
> > **kindly requesting a reorganization**
> >
> > Could it be possible to reorganize your answer that addresses my questions in the same format as 1, 2, 3, 4; and for each point can you guide me where to look in the general response or copy the related part?
> >
> > If not I'll do my best, but in this format there are quotes without full context and it will be hard to respond everything at once. I had to re-iterate my questions, and the followup discussions can be inefficient with this way.

---

> > > ### Author Response · Authors · 2023-11-21
> > >
> > > Thanks for the response!
> > >
> > > We have restructured our response to highlight which part corresponds to which of your concerns.
> > > About the general response, it basically highlights what changes were made to the draft and why. Moreover, it underlines our central thesis which might not have been clear before. Therefore, we request the reviewer to read it.

---

> ### Author Response · Authors · 2023-11-21
> **Continued**
>
> Next we address minor suggestions/questions:
>
> > Does the sparsity ratio changes from layer to layer of GPT-J?
>
> The sparsity ratio does not change significantly for $W_K$ and $W_Q$ but drops a little for $W_V$ with deeper layers. If we believe the weights that simulate GD to be sparse (as in Von Oswald et al. and Akyürek et al.), this contradicts the probing experiments in Akyürek et al. which suggest that the optimization happens in higher layers. We have changed our sparsity figure (**Figure 4**) to reflect this, and also updated the appendix with more details.
>
> > Why do you switch between models GPT-J vs LLama?
>
> While LLaMa is open-sourced, to our knowledge its intermediate training checkpoints are not released (which are used in Fig.3; notice that the x-axis is the checkpoint step). Therefore, we used a slightly smaller model (GPT-J, with ~6B parameters) with available checkpoints to compare the evolution of weights and ICL ability with training steps.
>
> > About SGD experiments:
>
> We did not change the batch size which was fixed to 2. The reason is that, like Von Oswald et al., the batch size was equal to the total samples available in context and for Akyürek et al., the batch size is 1 by construction. On the suggestion of the reviewer, we do show an ablation with batch size = 1 for SGD/Adam in Appendix A.
> We shuffled the samples in all experiments and also did multiple passes (assuming multiple layers perform serial GD in the model).

---

> ### Comment · Reviewer_uBmS · 2023-11-22
> **thank you!**
>
> **1) The GD is an ambiguous algorithm**
>
> The new version is much clear in terms of argumentation, thank you!. But now the main argument of the paper has been substantially changed or had been made it clear to me. So, I am re-reviewing.
>
> The argument the paper is refuting now is **"Hypothesis-2 => Hypothesis-1"** as written in the rebuttal. I am copying them here:
>
> - **H2**: For a given well-defined task, there exist Transformer weights such that ICL is algorithmically equivalent to GD (whole model or sub-model).
>
> - **H1**: For any Transformer weights resulting from self supervised pretraining and for any well-defined task, ICL is algorithmically equivalent to GD (whole model or sub-model).
>
> Let me come up with three more hypotheses for clarity
>
> - **H1O**: For linear Transformer resulting from meta training on the task family in H2, ICL is algorithmically equivalent to GD.
>
> - **H1A**: For large enough Transformer resulting from meta training on the task family in H2, ICL is algorithmically equivalent to Bayes optimal decision rule
>
> - **H1LM**: For an LM resulting from self-supervised pretraining on a large corpous, ICL is algorithmically equivalent to GD (whole model or submodel)
>
> **a)** I think "**H2 => H1**" argument **neither made/suggested** by Oswald et al. nor Akyurek et al.
>
> My understanding is that both papers shows **H2** by theoretical constructions, and Oswald shows **H1O** and Akyurek shows
> **H1A** by experimentation. I think what could be problematic in these papers, and I think this is what the experiments of this paper tries to refute **(H2, H1O/A) => H1LM**. This claim is mildy suggest in Oswald paper, but not in Akyurek (my point in (4)). So, I would **strongly** suggest going back to this (refuting **(H2, H1O/A) => H1LM**) positioning, which is the way I understood at the beginning, or please clarify further.
>
> **b)** To reiterate my worry **in the previous (1)**: **H1LM** is too strong as GD is an unspecified algorithm. What succefully refuteed by the experiments of this paper is ICL cannot be specific form of GD with cross entropy loss on intuitive subsets of LM --- not all forms of GD.
>
> So, overall what I delineate here still remains a major argumentation issue to me, I am happy to change my scores when this is fixed.
>
> **2) Thank you! This looks great!**
>
> **3) Looks great!**
>
> **4) Please see (1)**
>
> I really liked the new $ICL$ and $\hat{ICL}$ distinction in the notation.

---

> > ### Author Response · Authors · 2023-11-23
> >
> > We thank the reviewer for this detailed dive into our work.  Your feedback is super useful for making our work clearer.  Please take a look at the updated draft (we noted the changes in a new general response).  In response to your comments:
> >
> > **(a)**:
> > >I think "**H2 => H1**" argument neither made/suggested by Oswald et al. nor Akyurek et al.
> >
> > In the introduction, our hypothesis 2 now uses $\widehat{\text{ICL}}$ (previously, it was $\text{ICL}$, which may have caused confusion). Hypothesis 2 is about “in-context learning” that **does not emerge** from pretraining (called $\widehat{\text{ICL}}$ in our work). This covers both: (1) the transformers with hand-constructed weights from Akyurek et al. and Oswald et al. (**H2** in your words), for which no method is proposed to achieve _by training_, as well as (2) the transformers which are trained using the ICL objective (“meta training on the task family” in your words). In essence, our hypothesis 2 covers **H2** as well **H1O**/**H1A** from your rewritten hypotheses.
> >
> > > this paper tries to refute **(H2, H1O/A) => H1LM**.
> >
> > Yes and no! :) We are studying this implication and find no conclusive evidence in support of it.
> > A conclusive refutation would require more effort in exploring all the possible conditions.
> >
> > > This claim is mildy suggest in Oswald paper, but not in Akyurek
> >
> > Both these works (along with a growing literature) use “in-context learning” to refer to $\widehat{\text{ICL}}$, which is inaccurate because $\text{ICL}$ is widely understood as the emergent ability of transformers _trained on_ **task-agnostic** data.
> >
> > Using linear regression as the family of tasks under study for ICL is perfectly reasonable, but the transformer under study that develops this ability should not be trained _explicitly and exclusively_ for this family of tasks.
> >
> > **(b)**:
> >
> > > … H1LM is too strong as GD is an unspecified algorithm … What succefully refuteed ….
> >
> > We are casting doubt on the claim, but **not** refuting it (if you spot such strong claims in our paper, please let us know so we can fix it). We agree that we have conclusively shown that ICL can not be these specific forms of GD (on the whole model and the specified intuitive submodels). But as you already mentioned, doing this study for all possible sub-models and all forms of GD is computationally infeasible. In the future, we aim to study how to pinpoint these submodels efficiently and also find alternate functional explanations for ICL.

---

### Author Response · Authors · 2023-11-21
**General Response**

We thank all the reviewers for the constructive feedback.

Below is the list of the updates we have made in our draft to address the comments in the reviews.

**Major paper updates (updated contents are highlighted in purple):**

- We have removed the figure on the first page since that seemed to have caused confusion about our focus, and have revised the wording in the introduction to make our arguments clearer and more concrete.
- We have updated our background section (particularly section 2.1) to clarify its connection to section 3.
In section 3, we have included Fig.1 to show the confusing landscape of ICL vs GD hypothesis. After the feedback from reviewers, we have made changes to both section 3.1 and 3.2 to make our arguments precise.
- We have updated our section 4 to include experimental results about hypothesis 2 where GD is performed on the implicit sub-model. We thank reviewer uBmS for their suggestions.

**Appendix updates (updated contents are highlighted in purple):**

- In Appendix A, we have added more experimental studies pertinent to order sensitivity experiments with both versions of GD.
- In Appendix D, we have added more fine-grained analysis of sparsity in model weights.
- In Appendix E and F, we have added numerous other experimental results regarding the comparison of ${\text{ICL}}$ and ${\text{GD}}$/$\widehat{\text{GD}}$.

Furthermore, we would like to address a common confusion about the intent of our work:

**What is the main thesis of this work?** The gist of the message that we are trying to communicate is as follows:  Prior works (Dai et al., Akyürek et al., Von Oswald et al., among others) provide theoretical formalisms that directly speak to hypothesis 2 (expressive power of attention layer), which are nice! **However,** we are interested in the curious question of whether these results generalize to more realistic setups. An additional motivation here is that, when reporting the prior work’s results, the boundaries of these conclusions are stretched beyond the scope of their assumptions in ways that are closer to hypothesis 1. (***). Therefore, we inspect whether these findings extend beyond their assumptions.

For absolute clarity, we are **not claiming that the prior work is incorrect**

(***) **Is there any evidence that the prior conclusions are interpreted outside the scope of their assumptions?** Here are several concrete instances:
- Most works make repeated mentions of “${\text{ICL}}$” (an emergent phenomenon), even though what they actually mean is simulating $\widehat{\text{ICL}}$ by training a Transformer on task-specific data using _ICL objective_, which are quite different! We distinguish these two by defining two notations in Fig.1: “${\text{ICL}}$” (real-practice, pretraining on task-agnostic data) vs. $\widehat{\text{ICL}}$ (simulated via ${\text{ICL}}$ objective). Explained in detail in section 3.1.
- The conclusions of the prior work are often reported with incomplete context and assumptions. For example, the title of a published work is: “Language Models Implicitly Perform Gradient Descent as Meta-Optimizers” though a more accurate statement is “Language Models **Have the Capacity to** Implicitly Perform Gradient Descent as Meta-Optimizers”
- Dai et al. provide empirical results in terms of similar accuracy of ICL and GD to support their claim for hypothesis 1 (full-model equivalence of GD and ICL). However, as we argue theoretically and empirically, equivalence on accuracy does not imply functional equivalence. In our empirical analysis (section 4) we show that, while ICL and GD might lead to similar accuracy, they lead to very different distribution over tokens, which indicates their functional discrepancy.
- Stretching the conclusions is also permeated throughout the text of the papers (we provide examples in other responses). Furthermore, there is a growing body of works (Kwangjun et al. 2023; Li et al. 2023; Ren et al. 2023; Zhang et al; 2023; among others) that are adopting the setup or prior work (training Transformers with ${\text{ICL}}$ objective, aka $\widehat{\text{ICL}}$ in our notation) to make various conclusions about ICL, with an implicit assumption that any finding for $\widehat{\text{ICL}}$ will extend to ${\text{ICL}}$.

Again, **our intention is not to dismiss prior work.** These are excellent and innovative works. Still, we feel that the community in our academic discourse would benefit from additional clarification on the implications of these works, which we worry might be misunderstood.

We continue to appreciate your valuable feedback to make this work more accurate and clear.

---

> ### Author Response · Authors · 2023-11-21
> **References**
>
> - Ahn, Kwangjun, et al. "Transformers learn to implement preconditioned gradient descent for in-context learning." arXiv preprint arXiv:2306.00297 (2023).
> - Li, Shuai, et al. "The closeness of in-context learning and weight shifting for softmax regression." arXiv preprint arXiv:2304.13276 (2023).
> - Ren, Ruifeng, and Yong Liu. "In-context Learning with Transformer Is Really Equivalent to a Contrastive Learning Pattern." arXiv preprint arXiv:2310.13220 (2023).
> - Zhang, Ruiqi, Spencer Frei, and Peter L. Bartlett. "Trained Transformers Learn Linear Models In-Context." arXiv preprint arXiv:2306.09927 (2023).

---

### Author Response · Authors · 2023-11-23
**More revisions made to the draft!**

Thanks for your comments thus far! We have made additional changes to the draft:
- We have revised the statement of Hypothesis2 so that it uses $\widehat{\text{ICL}}$. Previously, we had “ICL” here since it was not defined in the intro, but this led to confusion about the meaning of this hypothesis. Now we use $\widehat{\text{ICL}}$ in Hyp2 and define what it means in the text (models that do not result from self-supervised pre-training).
- Based on the reviewer’s feedback, we reorganized section 3 to make the distinction between our study of limitations in previous works and our order-sensitivity analysis clearer.
- Revised the figures for more font consistency and clarity. In particular, notice the changes to the figure1 and its caption.

We ask the reviewers who have not commented yet to please share their feedback with us in the remaining hours of the discussion period.

As before, please continue to give us feedback to make this work more accurate and clear. 🙏

---

### Meta-Review · Area_Chair_cSSs · 2023-12-05

**Metareview:**

This paper takes a critical look at certain claims in the ICL literature that “ICL works by Gradient Descent”, which sometimes conflate representability and learnability. The authors emphasize that many prior works are statements about representation, which do not necessarily extend to the actual behavior of trained models. As evidence of this separation, they show that experimentally, ICL is not invariant to the order of examples in certain settings (whereas GD is). They also articulate several heuristic concerns about the assumptions in prior works.

Reviewers generally agree with the message of the paper. Their primary concerns are:
1. Novelty: It is not clear whether this perspective is new & insightful to the community. While it is true that certain papers have over-claimed, many in the community are quite aware of the limitations of these prior works (including the present reviewers).
2. Technical soundness: Many claims made in the present paper are informal and imprecise, and moreover not supported by evidence. Precision is especially important in a paper that aims to refine inaccuracies in prior works. Concretely, for example, Hypothesis 1 and 2 are not well-defined: what does “GD (whole model or sub-model)” mean formally? The details matter a lot, as the authors acknowledged in their rebuttal. Similarly, the remarks about linear regression are not supported, as Reviewer pVVn noted.

The primary strength of this paper is the experiments on order-stability, which are interesting and new.

The authors made significant revisions during the rebuttal period to partially address the concerns. Their revisions are appreciated and improved the quality of the paper.
However, no reviewer strongly supported the paper even after revision, and after reading the paper myself, the significant concerns above were not adequately addressed.
Thus I must recommend rejection.

I encourage the authors to revise their work and submit to a future venue. The experiments are interesting, and could be the basis of a strong paper, if the claims about them are made more precise and cohesive. I also suggest the authors de-emphasize objections to prior work unless they can be made formally.

**Justification For Why Not Higher Score:**

The concerns on framing, precision, and novelty were significant enough to warrant rejection.

**Justification For Why Not Lower Score:**

N/A

---

### Decision · Program_Chairs · 2024-01-16

Reject